# CardioBench: Do Echocardiography Foundation Models Generalize Beyond the Lab?

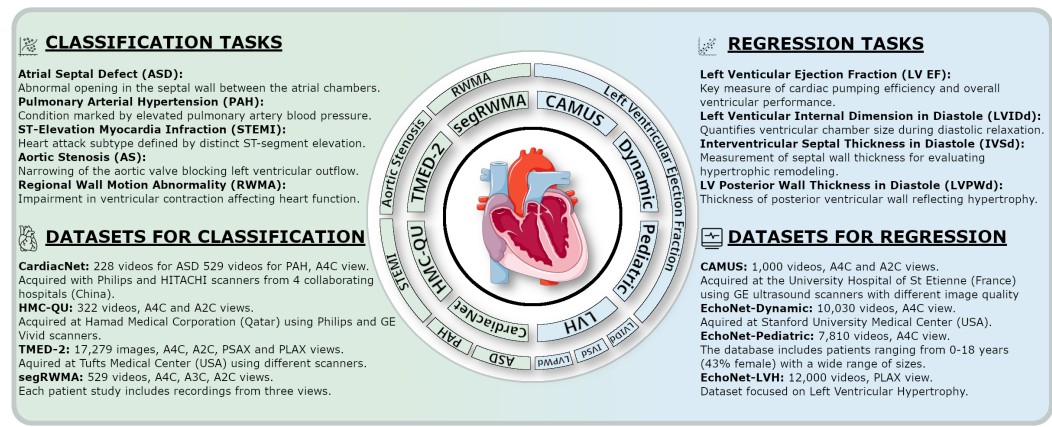

Figure 1: CardioBench is a standardized benchmark unifying 8 datasets, covering 4 regression tasks and 5 classification tasks across multi-view echocardiography.

## Abstract

Foundation models (FMs) are reshaping medical imaging, yet their application in echocardiography remains limited. While several echocardiography-specific FMs have recently been introduced, no standardized benchmark exists to evaluate them. Echocardiography poses unique challenges, including noisy acquisitions, high frame redundancy, and limited public datasets. Most existing solutions evaluate on private data, restricting comparability. To address this, we introduce CardioBench, a comprehensive benchmark for echocardiography FMs. CardioBench unifies eight publicly available datasets into a standardized suite spanning four regression and five classification tasks, covering functional, structural, diagnostic, and view recognition endpoints. We evaluate several leading FM, including cardiac-specific, biomedical, and general-purpose encoders, under consistent zero-shot, probing, and alignment protocols. Our results highlight complementary strengths across model families: temporal modeling is critical for functional regression, retrieval provides robustness under distribution shift, and domain-specific text encoders capture physiologically meaningful axes. General-purpose encoders transfer strongly and often close the gap with probing, but struggle with fine-grained distinctions like view classification and subtle pathology recognition. By releasing preprocessing, splits, and public evaluation pipelines, CardioBench establishes a reproducible reference point and offers actionable insights to guide the design of future echocardiography foundation models.

## 1 Introduction

Foundation models (FMs) have become a transformative force in vision and language domains, demonstrating remarkable capabilities across diverse tasks, including zero-shot image classification and retrieval (Radford et al. (2021); Jia et al. (2021)), visual grounding and segmentation (Ghiasi et al. (2022); Li et al. (2022)), and multimodal reasoning (Singh et al. (2022); Alayrac et al. (2022)).

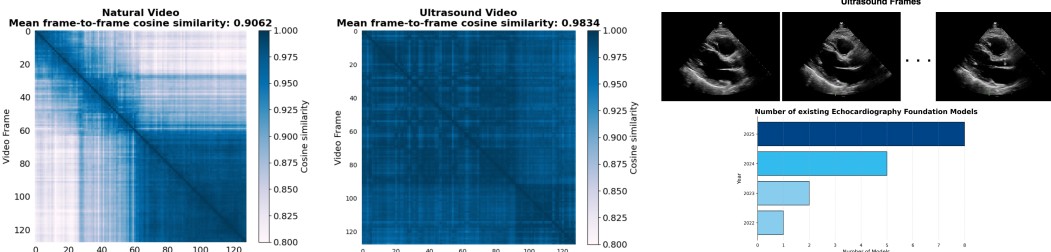

Figure 2: The figure on the left shows frame-level cosine similarity matrices: natural video frames from the SumMe dataset (Gygli et al. (2014)) versus echocardiography video frames extracted using SigLIP2 (Tschannen et al. (2025)). Echocardiography videos exhibit much higher frame-to-frame similarity compared to natural videos, making informative feature extraction more challenging. The figure on the right illustrates the number of echocardiography foundation models released each year: by mid-2025, there are 8 models published.

Large-scale architectures such as CLIP, DINOv3, and SigLIP2 demonstrate that self-supervised and multimodal learning produce general-purpose backbones with strong transferability across downstream tasks (Radford et al. (2021); Siméoni et al. (2025); Tschannen et al. (2025)). Similarly, in medical imaging, large-scale pre-training has been shown to improve generalization across tasks. For 2D radiography, the abundance of public datasets has enabled FMs to advance disease classification and localization (Irvin et al. (2019); Johnson et al. (2019)), while for 3D data, several architectures have achieved state-of-the-art segmentation and detection results (Roy et al. (2023); Huang et al. (2023)).

While foundation models in medical imaging have achieved notable progress in 2D and 3D modalities, this success has largely been driven by the availability of large, standardized datasets. Ultrasound, and especially echocardiography, poses unique challenges as a temporal sequence of 2D images, with public datasets limited both in scale and in the diversity of available video data. Additionally, ultrasound images are inherently noisy and temporally complex, with high frame-to-frame similarity that complicates effective representation learning (Kang et al. (2024); Song et al. (2024)). As illustrated in Figure 2, ultrasound videos exhibit a higher mean frame-to-frame cosine similarity compared to natural videos, reflecting the low signal-to-noise ratio and limited visual diversity of the modality. These traits have been linked to reduced robustness and limited generalization when training models directly on noisy images (Javed et al. (2024)). Despite these challenges, there is growing interest in developing ultrasound foundation models, as evidenced by the increasing number of models proposed each year (Figure 2). However, most of these models have been developed and evaluated on private datasets, which makes it difficult to assess their generalizability. This fragmentation hinders progress and creates an urgent need for a standardized evaluation protocol to provide a common ground for fair comparison and benchmarking.

Furthermore, it remains unclear how these modality-specific models compare to general-purpose vision foundation models, which have much larger diversity in training data. This raises several fundamental questions: How do echocardiography foundation models perform relative to each other under a fixed evaluation protocol? Are their learned representation spaces fundamentally different from those of general-purpose models, and how do these differences affect downstream tasks? To what extent can they enable zero-shot transfer, and do they exhibit systematic biases across datasets or clinical tasks? Addressing these open questions is essential for establishing reliable foundations for echocardiography AI, with direct implications for both methodological progress and the safe and reliable translation of these technologies into clinical practice.

This work introduces **CardioBench** (see Figure 1), a comprehensive benchmark for echocardiography foundation models. By unifying eight publicly available datasets into a standardized evaluation suite spanning four regression and five classification tasks, CardioBench establishes the common ground for fair, reproducible, and clinically meaningful comparison. Unlike prior efforts that focused on individual datasets or tasks, CardioBench enables systematic evaluation across functional and structural endpoints, providing a robust basis for tracking progress in this emerging field. It compares leading cardiac-specific models against general-purpose vision and biomedical encoders

under consistent zero-shot, probing, and alignment protocols, offering controlled analysis of how architectural design, temporal modeling, and supervision strategies shape transferability. To maximize accessibility and reproducibility, CardioBench provides standardized dataset preprocessing and data splits together with unified evaluation scripts, ensuring that results are directly comparable and easily extendable by the community.

Beyond results, CardioBench provides actionable insights into what drives performance in echocardiography foundation models: the role of temporal modeling, the importance of text encoders, the robustness of retrieval-based methods, and the surprising strengths and weaknesses of generalist backbones. We expect CardioBench to: (1) stimulate the development of new models tailored to the unique challenges of echocardiography, (2) establish a systematic way of measuring model quality for scientific progress, and (3) guide future pretraining strategies by revealing which architectural and supervision choices yield meaningful representations.

## 2 RELATED WORK

Recent works have advanced benchmarking and foundation models in medical imaging across multiple domains. Bassi et al. (2024) builds a large-scale segmentation benchmark across nine abdominal organs to test the models under distribution shift. Beyond performance, Jin et al. (2024) emphasized fairness by assessing foundation models across multiple modalities and sensitive attributes. At the same time, Huix et al. (2024) highlights the difficulty of transferring general-purpose FMs to specialized modalities. In echocardiography, M Alaa et al. (2022) provided an important early benchmark by assembling four public datasets into 31 tasks, establishing the first standardized protocol for model comparison. Many of these tasks, however, overlap across datasets and views, offering breadth but less diversity in evaluation. Since then, several echocardiography foundation models have been released, many of which are evaluated only on private datasets, which limits reproducibility and fair comparison across methods (Song et al. (2024)). Together, these works highlight the absence of a standardized benchmark for echocardiography foundation models, underscoring the need for a public protocol that enables fair evaluation under noise and domain shifts in cardiac ultrasound.

## 3 BENCHMARKING

### 3.1 CLINICAL TASKS AND DATASETS

Echocardiography offers a complete view of the heart, capturing its motion, structure, and pathological states across time. Unlike prior work, such as ETAB M Alaa et al. (2022), **CardioBench** is designed to benchmark recently developed echocardiography foundation models, introducing a more diverse set of clinically relevant endpoints and datasets, enabling fair and reproducible evaluation. To rigorously benchmark foundation models in this domain, we design tasks that capture functional, structural, and diagnostic aspects of clinical practice, as illustrated in Figure 1 (see Appendix C for details on datasets used). Functional tasks reflect the heart's movement over time, with Left Ventricular Ejection Fraction (LV EF) regression serving as a standard measure of global cardiac performance that requires models to capture temporal dynamics across the cardiac cycle. Structural tasks emphasize the anatomical properties of the heart, targeting diastolic measurements (IVSd, LVIDd, LVPWd) to assess the spatial localization of cardiac walls. At the same time, diagnostic tasks focus on disease classification, including aortic stenosis (AS), pulmonary arterial hypertension (PAH), atrial septal defect (ASD), ST-elevation myocardial infarction (STEMI), and regional wall motion abnormality (RWMA) from 3 different views, thereby testing adaptability to diverse clinical targets.

Beyond core tasks, the CardioBench also accounts for echocardiography's broader context, including its multi-view nature and potential demographic biases. Echocardiography is inherently multi-view, with different pathologies and anatomical structures visible only from specific perspectives. View classification is therefore essential, as accurate recognition enables physicians to interpret the correct structures and ensures that automated models apply the appropriate downstream diagnostic tasks. We additionally analyze demographic and acquisition-related factors, providing insight into subgroup robustness.

## 3.2 MODELS

For evaluation, we consider three categories of foundation models: those designed specifically for echocardiography, those trained on broader biomedical data, and large-scale general-purpose models. These span a wide range of architectural choices, from models without text supervision to those with temporal transformers over frame sequences or purely image-level extractors. Taken together, these variations in scale, architecture, and pretraining strategy allow us to assess how different design choices transfer to echocardiography interpretation (see Appendix B).

**Echocardiography–specific FM.** We evaluate the four Echocardiography foundation models with publicly released weights available at the time of writing. The earliest, EchoCLIP (Christensen et al. (2023)), introduced a vision–language approach to cardiac ultrasound. EchoPrime (Vukadinovic et al. (2024)) built on this idea with a stronger video encoder and a larger dataset, while also incorporating a separate view classifier and relying on report retrieval at inference time. In parallel, PanEcho (Holste et al. (2025)) explored an alternative direction by discarding text supervision and instead combining frame features with temporal aggregation in a multitask setup, while EchoFM (Kim et al. (2024)) explored a generative pretraining strategy centered on reconstructing cardiac motion.

**Biomedical and general-purpose FM.** To assess transfer from broader domains, we also include BioMedCLIP (Zhang et al. (2023)), a vision–language model pretrained on millions of biomedical image–text pairs spanning radiology, microscopy, pathology, and ultrasound. For comparison, we evaluate two large-scale general-purpose models trained at internet scale: DINOv3 (Siméoni et al. (2025)), a self-supervised vision transformer, and SigLIP2 (Tschannen et al. (2025)), a vision–language model aimed at producing stronger dense representations. Together, these models enable testing of how far biomedical and generic pretraining can transfer to echocardiography tasks, and whether domain-specific pretraining is required to achieve strong performance.

## 4 EXPERIMENTS

We design experiments to examine two complementary aspects of foundation models: **(i) the capacity to perform clinically relevant tasks without task-specific training**, and **(ii) the quality of their learned representations for downstream adaptation**. Therefore, we focus on zero-shot evaluation and probing, while excluding fine-tuning and few-shot training, as both are prone to overfitting and require substantial labeled data for stable performance (Silva-Rodriguez et al. (2024)). Further details on zero-shot evaluation, prompt design, and probing implementations are provided in Appendices D and E.

Foundation models are evaluated on both predictive accuracy and the structure of their learned representations. We therefore report metrics across four dimensions: task performance, clustering consistency, cross-modal alignment, and demographic robustness. For task performance, we use Mean Absolute Error (MAE) as the primary regression metric and macro-averaged F1 for classification and view classification, with additional measures reported in the Appendix G. Clustering consistency is assessed using the Adjusted Rand Index (ARI), which measures how well embedding clusters recover ground-truth echocardiography views. Cross-modal alignment is evaluated by testing whether visual embeddings align with text prompts. Finally, demographic robustness is examined through subgroup analyses of EF errors stratified by sex, age, BMI, and image quality, with complete subgroup tables provided in Appendix H

### 4.1 RESULTS

We summarize the performance of selected models in a zero-shot setting in Table 1. PanEcho is the most consistent performer, achieving the best and second-best results for ejection fraction (EF) estimation on EchoNet-Dynamic and EchoNet-Pediatric, and consistently outperforming all competitors on the structural regression tasks from EchoNet-LVH. Its strength also extends to classification, where it achieves the highest score of 58.90% on TMED-2 aortic stenosis (AS) detection. EchoPrime shows strong results in both regression and classification tasks, which is particularly interesting given its retrieval-based inference framework and the potential influence of similarities between test cases and its private database.

Table 1: Zero-shot results across 4 regression tasks and 5 classification tasks on 8 publicly available datasets. Models with video-based training are marked with ●. Regression performance is reported in **MAE**, while classification is reported in **F1-macro** score. Blue columns are regression tasks, while green columns are classification tasks. The best results are shown in **bold**, and the second best are underlined.

| Model | Dynamic | CAMUS | Pediatric | LVH | | | CardiacNet | | HMC-QU | TMED-2 | segRWMA | | |
| --- | --- | --- | --- | --- | --- | --- | --- | --- | --- | --- | --- | --- | --- |
| | LV EF | | | LVIDd | IVSd | LVPWd | ASD | PAH | STEMI | AS | A2C | A3C | A4C |
| **EchoCLIP** Christensen et al. (2023) | 9.99 | 9.83 | 13.80 | 0.79 | 0.57 | 0.41 | **47.88** | **46.96** | **52.51** | 44.13 | 35.68 | 36.27 | 14.29 |
| **EchoPrime** Vukadinovic et al. (2024) ● | 7.78 | 14.00 | **5.44** | - | - | - | - | - | - | 44.13 | - | - | - |
| **PanEcho** Holste et al. (2025) ● | **5.79** | 11.63 | 9.10 | **0.36** | **0.21** | **0.18** | - | - | - | **58.90** | 30.50 | 24.30 | 20.52 |
| **BioMedCLIP** Zhang et al. (2023) | 13.83 | 18.87 | 18.30 | 0.97 | 0.28 | 0.26 | 40.24 | 25.75 | 33.33 | 44.13 | 37.66 | 32.10 | 6.67 |
| **DINOv3** Siméoni et al. (2025) | 14.67 | 9.88 | 18.24 | 0.69 | 0.28 | 0.22 | 36.49 | 41.44 | 34.21 | 44.13 | **47.83** | 48.00 | **48.15** |
| **SigLIP2** Tschannen et al. (2025) | 14.66 | **9.28** | 18.22 | 0.69 | 0.28 | 0.22 | 36.49 | 24.11 | 32.43 | 17.38 | 47.25 | **72.02** | 47.17 |

Table 2: Linear probing results across 3 regression tasks and 4 classification tasks on 4 publicly available datasets. Regression performance is reported in **MAE**, while classification is reported in **F1-macro** score. Reported $\Delta$ values indicate absolute change relative to zero-shot. Models with video-based training are marked with ●. Blue columns are regression tasks, while green columns are classification tasks. The best results are shown in **bold**, and the second best are underlined.

| Model | LVH | | | | | | CardiacNet | | | | HMC-QU | | segRWMA | | | | | |
| --- | --- | --- | --- | --- | --- | --- | --- | --- | --- | --- | --- | --- | --- | --- | --- | --- | --- | --- |
| | LVIDd | $\Delta$ | IVSd | $\Delta$ | LVPWd | $\Delta$ | ASD | $\Delta$ | PAH | $\Delta$ | STEMI | $\Delta$ | A2C | $\Delta$ | A3C | $\Delta$ | A4C | $\Delta$ |
| **EchoCLIP** Christensen et al. (2023) | 0.47 | 0.32 | 0.28 | 0.29 | 0.22 | 0.19 | 38.49 | 9.39 | 41.44 | 5.52 | 73.99 | 21.48 | 47.83 | 12.15 | 48.00 | 11.73 | 48.15 | 38.86 |
| **EchoPrime** Vukadinovic et al. (2024) ● | 0.41 | – | 0.25 | – | 0.19 | – | 52.66 | – | **63.36** | – | **80.00** | – | 8.33 | – | **68.48** | – | 48.15 | – |
| **PanEcho** Holste et al. (2025) ● | **0.35** | 0.01 | **0.18** | 0.03 | **0.15** | 0.03 | 58.53 | – | 61.51 | – | | | **72.73** | 42.23 | 47.47 | 23.17 | **64.78** | 44.26 |
| **EchoFM** Kim et al. (2024) | 0.57 | – | 0.32 | – | 0.24 | – | 50.48 | – | 41.44 | – | 71.82 | – | 47.83 | – | 48.00 | – | 48.15 | – |
| **BioMedCLIP** Zhang et al. (2023) | 0.52 | 0.45 | 0.30 | 0.02 | 0.23 | 0.03 | 58.53 | 1.20 | 41.44 | 15.69 | 55.44 | 22.11 | 47.83 | 10.17 | 48.00 | 15.90 | 48.15 | 41.48 |
| **DINOv3** Siméoni et al. (2025) | 0.47 | 0.22 | 0.28 | 0.00 | 0.21 | 0.01 | 56.76 | 22.36 | 58.85 | 17.41 | 75.00 | 40.79 | 47.83 | 0.00 | 48.00 | 0.00 | 48.15 | 0.00 |
| **SigLIP2** Tschannen et al. (2025) | 0.51 | 0.18 | 0.30 | 0.02 | 0.23 | 0.01 | **68.49** | 32.00 | 47.96 | 23.85 | 75.00 | 42.57 | 47.83 | 0.48 | 48.00 | 24.02 | 48.15 | 0.98 |

A notable observation is the performance of general-purpose foundation models such as SigLIP2 and DINOv3, which deliver strong results despite lacking cardiac-specific pretraining. SigLIP2, in particular, surpasses several specialized echocardiography models on CAMUS EF estimation and achieves competitive performance on segRWMA regional wall abnormality detection. At the same time, both SigLIP2 and DINOv3 perform nearly on par with PanEcho on EchoNet-LVH regression LVPWd. In classification, they achieve the highest scores in RWMA detection across all three views, even outperforming EchoCLIP, despite EchoCLIP being explicitly trained on cardiac ultrasound. This underperformance is most pronounced on the A4C view, where EchoCLIP lags by more than 34%. Nevertheless, EchoCLIP remains strong on several tasks, achieving F1 scores of 47.88% on ASD and 46.96% on PAH, surpassing the best general-purpose models by margins of 7.61% and 5.52%, respectively. On STEMI detection, it reaches 52.51%, representing an improvement of 18.3% over competitors.

The linear probing performance is summarized in Table 2. On regression tasks, PanEcho maintains a clear advantage, achieving the lowest errors across all EchoNet-LVH measurements (MAE of 0.35 on LVIDd, 0.15 on IVSd, and 0.30 on LVPWd), with only marginal improvements from linear probing ($\Delta \leq 0.03$). By contrast, general-purpose encoders such as DINOv3 and SigLIP2 show larger reductions in error (0.20–0.23 MAE), narrowing the gap to PanEcho, though they remain behind. These results illustrate that EchoNet-LVH structural regression benefits less from probing. For classification, linear probing yields more pronounced changes. SigLIP2 improves by 32% on ASD to reach 68.49% F1, outperforming all specialized models by nearly 10%. On PAH and STEMI, however, EchoPrime delivers the strongest performance, achieving 63.36% and 80.00%, while SigLIP2 remains competitive at 47.96% and 72.57%, respectively. These results show that general-purpose encoders can not only close the gap but, in some cases, even surpass specialized models.

In RWMA detection, PanEcho achieves the highest gains, with improvements of 42.23% on A2C and 44.26% on A4C, reaching 72.73% and 64.78%, respectively. EchoPrime excels on A3C, where it reaches 68.48%, while EchoCLIP remains flat at 48.00% across all views, converging with DINOv3 and SigLIP2 despite its cardiac-specific training. Overall, linear probing highlights complementary strengths with PanEcho remaining unrivaled on regression and two RWMA views, EchoPrime achieving the best results on PAH and STEMI, and SigLIP2 surpassing all competitors on ASD.

View classification results in Table 3 show that EchoPrime achieves the highest F1 scores on the majority of datasets, benefiting from its supervised, pretrained view classifier rather than relying

Table 3: View classification results across 8 publicly available datasets, reported in **F1-macro** score. Multi-view datasets are marked with ●. The best results are shown in **bold**, and the second best are underlined.

| Model | LVH | CardiacNet | CAMUS ● | Dynamic ● | Pediatric ● | HMC-QU ● | TMED-2 ● | segRWMA ● |
|---|---|---|---|---|---|---|---|---|
| **EchoCLIP** Christensen et al. (2023) | 1.76 | 27.12 | 33.11 | 8.55 | 20.95 | 34.33 | 14.25 | 16.86 |
| **EchoPrime** Vukadinovic et al. (2024) | **98.66** | 34.59 | 16.39 | **98.49** | **79.53** | **88.19** | **62.86** | 15.79 |
| **BioMedCLIP** Zhang et al. (2023) | 0.57 | **76.11** | 17.02 | 26.37 | 18.41 | 47.67 | 21.98 | **18.41** |
| **DINOv3** Siméoni et al. (2025) | 0.00 | 0.00 | 0.00 | 0.31 | 35.82 | 0.00 | 4.89 | 0.00 |
| **SigLIP2** Tschannen et al. (2025) | 29.05 | 8.75 | **57.01** | 87.29 | 45.32 | 41.37 | 16.17 | 2.43 |

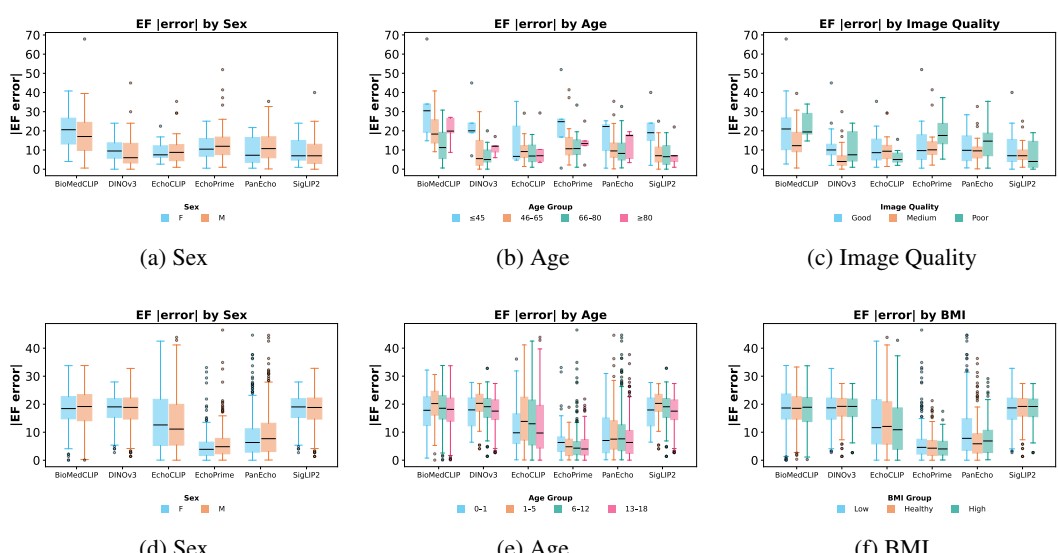

(a) Sex          (b) Age          (c) Image Quality

(d) Sex          (e) Age          (f) BMI

Figure 3: Absolute EF error distributions across demographic subgroups in **CAMUS** (a–c) and **EchoNet-Pediatric** (d–f).

solely on text–prompt alignment. It leads on five out of eight datasets, highlighting the strength of its dedicated view recognition module. Interestingly, the remaining datasets are topped by models without cardiac-specific pretraining: BioMedCLIP achieves the best results on CardiacNet (76.11%) and TMED-2 (62.86%), while SigLIP2 outperforms all others on CAMUS (57.01%). By contrast, EchoCLIP, despite being trained specifically on echocardiography, fails to dominate on any dataset and often lags behind BioMedCLIP or general-purpose models. These findings suggest that while supervised view classifiers provide a clear advantage, large-scale pretraining on diverse medical or natural images can transfer surprisingly well to echocardiography view classification.

Subgroup analyses reveal distinct biases in EF estimation on CAMUS that are less pronounced in EchoNet-Pediatric (Figure 3), despite overall performance trends being consistent across models. On CAMUS (Figure 3a–c), subgroup differences are evident: younger patients ($\leq 45$) and scans labeled as "Good" quality show larger errors and wider spreads, likely reflecting distribution biases since most samples fall into the "Medium" quality category, where models perform best. A modest sex gap is also visible, with females showing slightly higher errors, particularly for EchoPrime and PanEcho. In the larger EchoNet-Pediatric cohort (Figure 3d–f), these disparities are less pronounced. Sex- and age-related differences largely disappear, while BMI exhibits the expected trend: healthy ranges yield lower errors, whereas both low and high extremes increase variability, consistent with the physics of ultrasound imaging, where excessive or insufficient tissue layers can degrade acoustic penetration and image quality. Across both datasets, SigLIP2 and DINOv3 maintain the most stable performance across demographic and acquisition subgroups, showing narrow error distributions and minimal subgroup-related shifts. BioMedCLIP, while consistently higher in absolute error, also shows relatively uniform behavior across subgroups. By contrast, PanEcho and EchoPrime demonstrate more outliers and wider error distributions across several subgroups, particularly in females and younger patients on CAMUS and in BMI extremes on EchoNet-Pediatric.

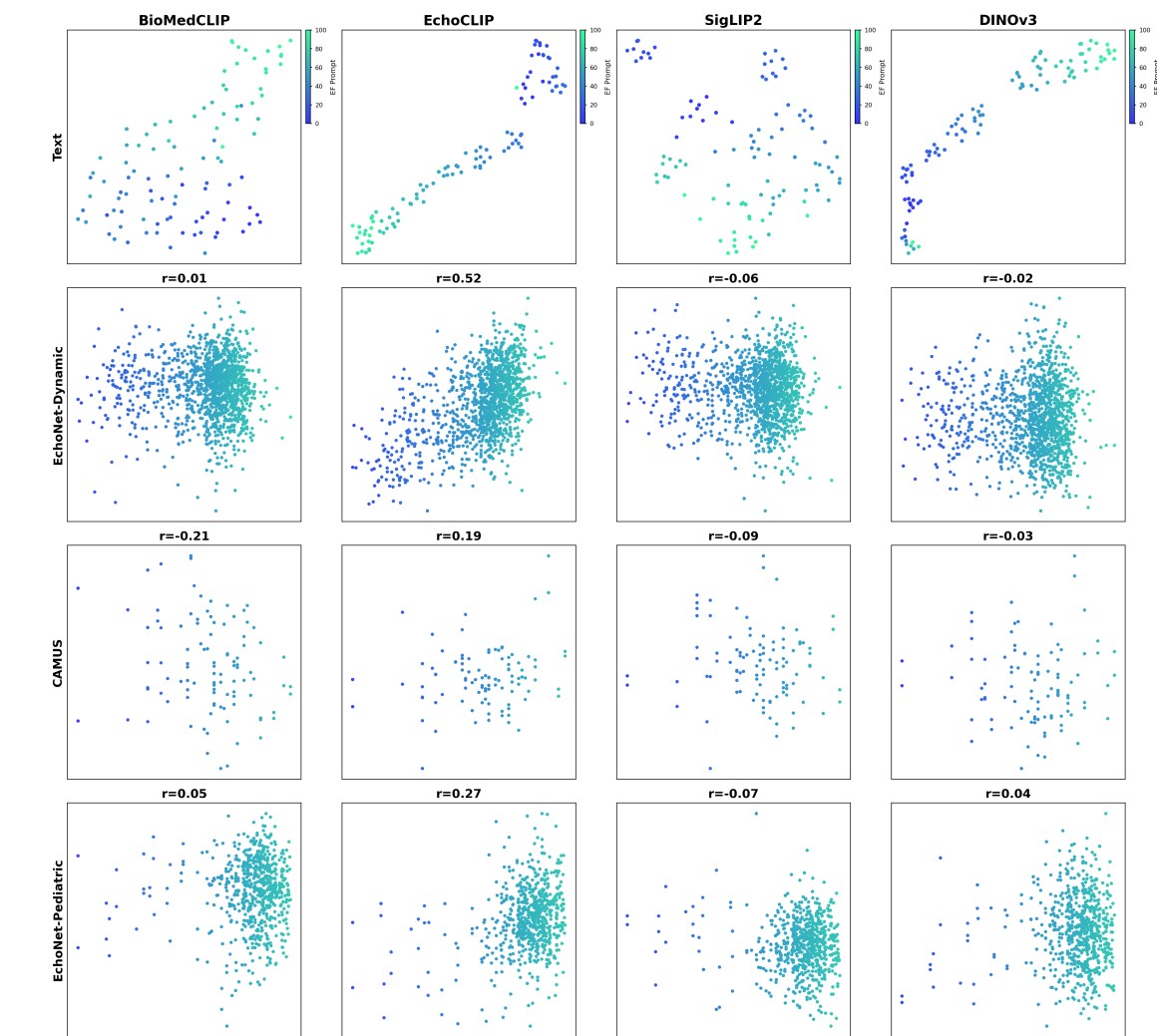

Figure 4: Top row: EF text prompt embeddings projected into 2D. Rows 2–4: alignment of visual embeddings with the EF text axis for each dataset.

## 5 DISCUSSION

CardioBench reveals that no single foundation model dominates across all tasks, datasets, and evaluation regimes. Instead, performance depends strongly on the interaction between model design choices, dataset characteristics, and evaluation setup.

**Modeling EF regression.** PanEcho and EchoPrime stand apart from the contrastive approaches in CardioBench because their zero-shot predictions are not driven by text encoders. PanEcho leverages its multitask design to achieve the lowest errors on EchoNet-Dynamic and strong results on Pediatric, showing that supervised EF knowledge can transfer effectively across datasets. EchoPrime, in contrast, benefits from retrieval: rather than modeling EF as a smooth continuum, it assigns labels by matching test cases to similar exemplars in its joint space. This discrete matching helps on EchoNet-Pediatric, where it outperforms contrastive models, but the approach fails on CAMUS, where scanner heterogeneity may distort embeddings and make nearest-neighbor matches unreliable. Both models incorporate temporal dynamics, but differ in how strongly their predictions depend on them. A frame-shuffling stress test on EchoNet-Dynamic (Appendix G, Table 7) demonstrates the contrast: PanEcho degrades when temporal coherence is removed, whereas EchoPrime remains relatively stable, suggesting that its retrieval mechanism can fall back on exemplar similarity even when sequence order is disrupted.

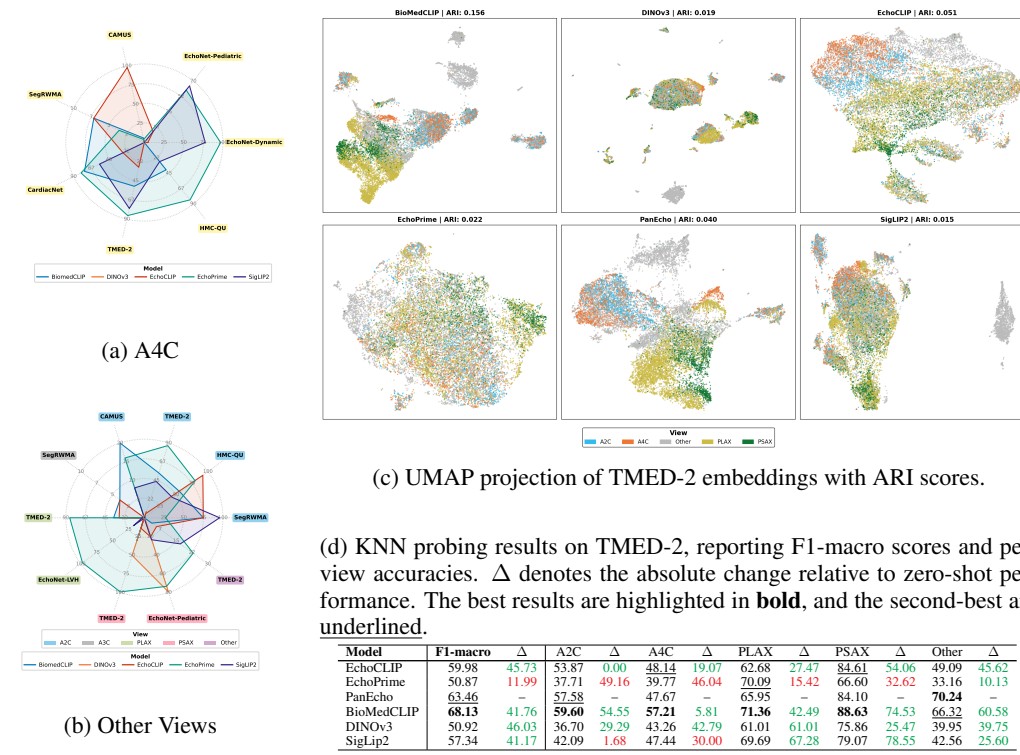

(a) A4C

(b) Other Views

(c) UMAP projection of TMED-2 embeddings with ARI scores.

(d) KNN probing results on TMED-2, reporting F1-macro scores and per-view accuracies. Δ denotes the absolute change relative to zero-shot performance. The best results are highlighted in **bold**, and the second-best are underlined.

| Model | F1-macro | Δ | A2C | Δ | A4C | Δ | PLAX | Δ | PSAX | Δ | Other | Δ |
|---|---|---|---|---|---|---|---|---|---|---|---|---|
| EchoCLIP | 59.98 | 45.73 | 53.87 | 0.00 | 48.14 | 19.07 | 62.68 | 27.47 | 84.61 | 54.06 | 49.09 | 45.62 |
| EchoPrime | 50.87 | 11.99 | 37.71 | 49.16 | 39.77 | 46.04 | 70.09 | 15.42 | 66.60 | 32.62 | 33.16 | 10.13 |
| PanEcho | 63.46 | – | 57.58 | – | 47.67 | – | 65.95 | – | 84.10 | – | 70.24 | – |
| BioMedCLIP | **68.13** | 41.76 | **59.60** | 54.55 | **57.21** | 5.81 | **71.36** | 42.49 | **88.63** | 74.53 | 66.32 | 60.58 |
| DINOv3 | 50.92 | 46.03 | 36.70 | 29.29 | 43.26 | 42.79 | 61.01 | 61.01 | 75.86 | 25.47 | 39.95 | 39.75 |
| SigLip2 | 57.34 | 41.17 | 42.09 | 1.68 | 47.44 | 30.00 | 69.69 | 67.28 | 79.07 | 78.55 | 42.56 | 25.60 |

Figure 5: Left: Radar plots of view classification accuracy across datasets. Right: UMAP projection of TMED-2 embeddings with KNN probing results

To examine contrastive approaches, we directly assess whether they encode EF as a cross-modal dimension. We construct a text axis from prompts spanning 0–100% EF, normalize these embeddings, and extract the first principal component (Figure 4). Visual embeddings from test videos (Figure 17) are then projected onto this axis, and their Pearson correlation with ground-truth EF quantifies alignment. This analysis reveals significant differences between models. EchoCLIP, trained on cardiac ultrasound reports, is the only model to recover a physiologically meaningful EF axis ($r = 0.52$ on EchoNet-Dynamic, $r \approx 0.2$–$0.3$ on CAMUS and Pediatric), suggesting that domain-specific text encoders can enforce monotonic cross-modal structure. BioMedCLIP, despite pretraining on extensive biomedical corpora, shows almost no alignment ($r \approx 0$), indicating that general medical semantics are insufficient to ground EF as a continuous variable. General-purpose models such as SigLIP2 and DINOv3 also result in near-zero correlations, yet achieve their strongest results on CAMUS. At first glance, this might suggest robustness to acquisition shifts; however, a closer look indicates that these gains are not physiologically grounded. Specifically, we observe that SigLIP2 achieves lower MAE on images with poor quality compared to those of higher quality (Figure 3c), which is counterintuitive from a clinical perspective. This pattern suggests that the apparent success of generalist models on CAMUS reflects sensitivity to dataset-specific artifacts rather than meaningful encoding of EF, explaining their poor generalization outside this narrow setting.

**Clustering challenges in view classification.** A similar picture emerges in view classification, where architectural choices again dominate over text alignment. EchoPrime achieves the strongest results across multiple datasets by leveraging its supervised view head, demonstrating that explicitly modeling clinical structure can result in zero-shot advantages. By contrast, EchoCLIP struggles to generalize beyond A4C despite being trained on this view, because its contrastive objective emphasizes alignment with reports rather than enforcing consistent view identity. As a result, its embeddings entangle clinical content with anatomical cues, limiting transfer even on its main training view. Large-scale encoders such as BioMedCLIP and SigLIP2 occasionally outperform specialized models on datasets like EchoNet-Pediatric and CAMUS, but UMAP projections (Figure 5c) of TMED-2 embeddings reveal that none of the models form globally distinct view clusters. Interestingly, BioMedCLIP, EchoCLIP, and PanEcho, which were not explicitly trained for view

classification, tend to group PLAX and PSAX together while mixing A2C and A4C, as these views are indeed visually similar within short-axis and long-axis families. kNN probing (Table 5d) recovers some discriminative power, ranking BioMedCLIP highest, followed by PanEcho and EchoCLIP, while SigLIP2 surpasses EchoPrime when its supervised view classifier is removed. This shows that EchoPrime's advantage comes almost entirely from its explicit classifier head, while other models contain partial view information in their embeddings that kNN can recover locally, but which does not form globally distinct clusters or generalize consistently across datasets.

**Embedding structures for pathology tasks.** Within CardioBench, inspection of the embedding spaces for classification tasks evidences that zero-shot performance is constrained by weakly discriminative representation spaces. The UMAP visualizations in Figure 18, pathology-present and pathology-absent cases form partially separable but substantially overlapping clusters, with limited intra-class compactness and low silhouette scores across datasets. This indicates the limited prioritization of pathology-specific cues in current visual backbones, which tend instead to encode broader distributional features. The contrast with linear probing, showing substantially higher performance for BioMedCLIP and SigLIP2, further highlights that discriminative signals are present but not aligned with text prompts or directly accessible for zero-shot. These findings underscore the gap between latent signal and usable representation, emphasizing the need for models that organize clinical information more explicitly.

Taken together, CardioBench makes clear that progress in echocardiography foundation models cannot be measured by zero-shot performance alone. Across regression, classification, and view recognition, the benchmark reveals a consistent pattern: models contain latent clinical signal, but its accessibility depends heavily on architectural design, training supervision, and the stability of the embedding organization. This points to several practical directions. First, explicit supervision for core clinical axes such as EF or view classification proves more reliable than expecting them to emerge implicitly, suggesting that pretraining pipelines should integrate lightweight but structured supervision. Second, temporal modeling is indispensable for functional tasks, as demonstrated by PanEcho, while retrieval-based matching offers complementary robustness, motivating hybrid approaches that combine the strengths of both. Third, domain-specific text encoders, as in EchoCLIP, can enforce physiologically meaningful cross-modal structure, but their advantage is not stable, underscoring the need to broaden cardiac text corpora. Finally, the surprisingly strong performance of general-purpose encoders such as SigLIP2 and DINOv3 highlights both an opportunity and a limitation: scale and diversity alone can produce robust baselines under domain shift, yet these models fail to organize clinical signals in a way that supports fine-grained reasoning. This suggests that future cardiac foundation models should not discard generalist architectures, but rather adapt them through targeted supervision and domain grounding, bridging the gap between robustness and clinical fidelity.

## 6 CONCLUSION

CardioBench demonstrates that echocardiography foundation models must be assessed through multi-task, multi-dataset evaluation to capture their true capabilities. Model performance depends on design and supervision choices, which shape strengths in temporal dynamics, retrieval, and clinically grounded representations. Future advances will likely come from hybrid approaches that combine these complementary benefits. By providing a standardized, publicly available benchmark, CardioBench establishes a baseline for fair comparison and a platform for developing the next generation of clinically meaningful models.

## 7 REPRODUCIBILITY STATEMENT

Details are provided in Appendix D, and all code and resources are available at https://anonymous.4open.science/r/CardioBench/.

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
