## A ABBREVIATIONS

| | |
|---|---|
| **EF** | Ejection Fraction |
| **IVSd** | Interventricular Septal Thickness in Diastole |
| **LVIDd** | Left Ventricular Internal Diameter in Diastole |
| **LVPWd** | Left Ventricular Posterior Wall Thickness in Diastole |
| **A2C, A3C, A4C** | Apical 2-, 3-, and 4-Chamber Views |
| **PLAX** | Parasternal Long-Axis View |
| **PSAX** | Parasternal Short-Axis View |
| **STEMI** | ST-Elevation Myocardial Infarction |
| **AS** | Aortic Stenosis |
| **PAH** | Pulmonary Arterial Hypertension |
| **ASD** | Atrial Septal Defect |
| **RWMA** | Regional Wall Motion Abnormality |
| **MAE** | Mean Absolute Error |
| **MSE** | Mean Squared Error |
| **RMSE** | Root Mean Squared Error |
| **ARI** | Adjusted Rand Index |
| **kNN** | k-Nearest Neighbors |
| **FM** | Foundation Model |
| **Δ** | Difference between max and min subgroup performance |

## B MODELS

Table 4 provides a high-level comparison, while below each model is described in more detail. CardioBench compares echocardiography-specific, biomedical, and general-purpose foundation models. **EchoCLIP** Christensen et al. (2023) adapts a ConvNeXt-B vision encoder with a CLIP-style text tower, trained contrastively on 1M A4C echo videos and reports, aligning video embeddings with task-specific prompts at inference. **EchoPrime** Vukadinovic et al. (2024) combines a multi-view ViT (mViT) with BioMedBERT and uses retrieval, projecting test videos into a joint embedding space and predicting by matching to labeled exemplars. **PanEcho** Holste et al. (2025) employs a ConvNeXt-T backbone with a temporal frame transformer, trained on 1.2M multiview echo videos for multitask regression and classification. **EchoFM** Kim et al. (2024) uses a ViT-L/16 video encoder trained on 290K multiview echo videos to learn general embeddings optimized for probing. As the linear heads are not provided, and the model doesn't have the text encoder, zero-shot cannot be performed **BioMedCLIP** Zhang et al. (2023) pairs a ViT-B/16 with PubMedBERT, pretrained on 15M biomedical image–text pairs spanning radiology, pathology, microscopy, and ultrasound. **DINOv3** Siméoni et al. (2025) is a self-supervised ViT-L/16 trained on 1.7B natural images with an aligned text encoder, applied by encoding frames and pooling temporally before probing or computing similarity with handcrafted prompts. Finally, **SigLIP2** Tschannen et al. (2025) is a multilingual vision–language model with a ViT-B/16 backbone and transformer text tower, trained on 10B WebLI pairs. Together, these models allow us to assess how far both biomedical and large-scale generic supervision can be transferred to echocardiography tasks, and whether modality-specific pretraining is necessary to achieve competitive performance.

Table 4: Summary of evaluated foundation models with vision/text encoders, temporal design, and dataset scale.

| Model | Vision encoder | Text encoder | Temporal modeling | Training data |
|---|---|---|---|---|
| **EchoCLIP** | ConvNeXt-B | CLIPTextModel | – | 1.03M A4C echo videos + reports |
| **EchoPrime** | mViT | BioMedBERT | Video encoder | 12.1M multiview echo videos + reports |
| **PanEcho** | ConvNeXt-T | – | Frame Transformer | 1.2M multiview echo videos |
| **EchoFM** | ViT-L/16 | – | Video encoder | 290K multiview echo videos |
| **BioMedCLIP** | ViT-B/16 | PubMedBERT | – | 15M image–caption pairs |
| **DINOv3** | ViT-L/16 | – | – | 1.7B natural images |
| **SigLIP2** | ViT-B/16 | ViT-style tower | – | 10B WebLI images |

Table 5: Echocardiography datasets used in this study, with their source, accessibility, and modality.

| Dataset | Source | Availability | Data type |
|---|---|---|---|
| **EchoNet-Dynamic** Ouyang et al. (2020) | Stanford AIMI | Open download | Video |
| **EchoNet-Pediatric** Reddy et al. (2023) | Stanford AIMI | Open download | Video |
| **EchoNet-LVH** Duffy et al. (2022) | Stanford AIMI | Open download | Video |
| **SegRWMA** Liu et al. (2023) | Kaggle | Open download | Video |
| **CardiacNet** Yang et al. (2024) | Kaggle | Open download | Video |
| **CAMUS** Leclerc et al. (2019) | Université de Lyon | Open download | Video |
| **HMC-QU** Degerli et al. (2021) | Private | Upon request | Video |
| **TMED-2** Huang et al. (2022) | Private | Upon request | Image |

Table 6: Summary of dataset characteristics, including sizes, splits, and available labels. Datasets for which we adopt the official split are indicated with ●, for other datasets we define a custom split.

| Dataset | Size | Train/Val/Test | Labels Used | View |
|---|---|---|---|---|
| EchoNet-Dynamic ● | 10,030 videos | 7,465/1,288/1,277 | EF | A4C |
| EchoNet-Pediatric ● | 7,810 videos | 6,365/798/658 | Age, Sex, Weight, Height | A4C |
| EchoNet-LVH ● | 12,000 videos | 10,490/1,167/343 | IVSd, LVIDd, LVPWd | PLAX |
| SegRWMA | 529 videos | 221/152/156 | RWMA | A4C, A3C, A2C |
| CardiacNet-ASD | 228 videos | 158/23/47 | ASD | A4C |
| CardiacNet-PAH | 471 videos | 319/51/106 | PAH | A4C |
| CAMUS ● | 1,000 videos | 400/50/50 | EF, Sex, Age, Image Quality | A4C, A2C |
| HMC-QU | 322 videos | 227/45/50 | STEMI | A4C, A2C |
| TMED-2 ● | 17,270 images | 360/119/119 | AS | A4C, A2C, PSAX, PLAX |

## C  DATASETS

In this section, we motivate the choice of datasets for evaluation, provide the distribution of values in each dataset, and describe the splitting strategy.

### C.1  DATASET SELECTION

Because echocardiography involves sensitive patient information, the number and size of public datasets are limited. We use eight datasets that are either openly downloadable or available upon request. Table 5 summarizes their key characteristics.

Table 6 provides an overview of dataset sizes, experimental splits, and available labels. For the CAMUS and TMED-2 datasets, we report the total number of unique videos and images, with splits defined at the patient level. For the other datasets, we assume one video per patient. We also indicate the type of annotations provided and describe how the data were partitioned into training, validation, and testing sets. Where applicable, we additionally summarize the distribution of classes.

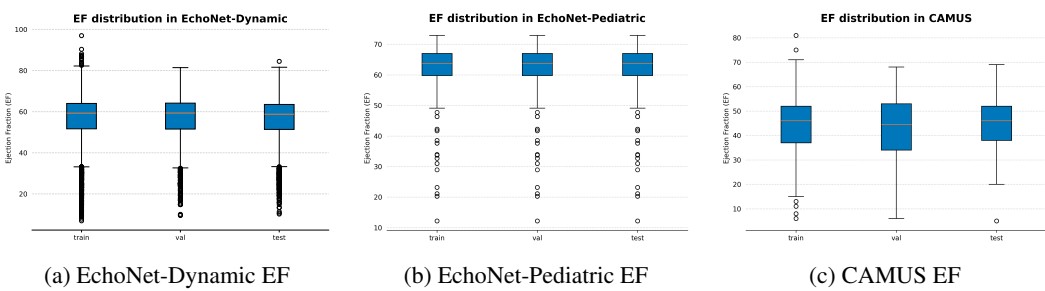

(a) EchoNet-Dynamic EF          (b) EchoNet-Pediatric EF          (c) CAMUS EF

Figure 6: Box plots of EF distributions across three datasets: EchoNet-Dynamic, EchoNet-Pediatric, and CAMUS.

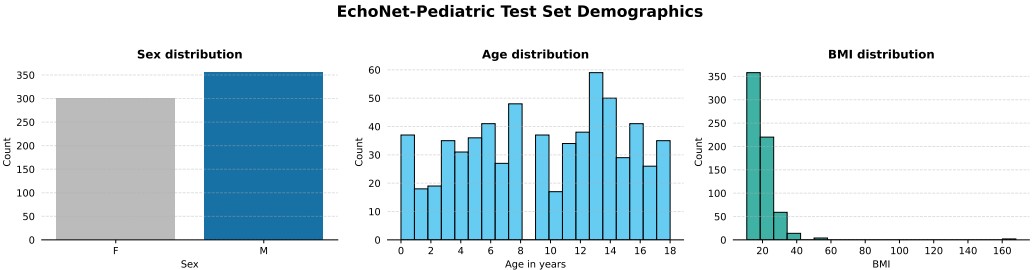

Figure 7: Distribution of sex, age, and BMI for video samples in the EchoNet-Pediatric dataset.

Figure 8: Distribution of sex, age, and image quality in the CAMUS dataset.

## C.2 DATASET DETAILS

**EchoNet-Dynamic.** The dataset consists of 10,030 A4C echocardiography videos, each from a unique patient. Every video is annotated with an EF value, with the distribution shown in Figure 6a.

**EchoNet-Pediatric.** The dataset comprises 7,810 videos, including 4,526 PSAX and 3,284 A4C echocardiography recordings, with one video per patient. Each video is annotated with EF, sex, age, weight, and height, from which body mass index (BMI) is derived. The EF distribution is shown in Figure 6b, and the demographic distributions are presented in Figure 7.

**EchoNet-LVH.** The EchoNet-LVH dataset contains 12,000 PLAX-view videos, each annotated with the frame on which structural measurements (IVSd, LVIDd, LVPWd) are performed, with their distributions shown in Figure 9.

**CAMUS.** The CAMUS dataset comprises 500 patients, each with two echocardiography views (A2C and A4C). Each video is annotated with sex, age, EF, and image quality. We follow the official split of 400 patients for training, 50 for validation, and 50 for testing. The EF distribution is shown in Figure 6c, and the demographic distributions are presented in Figure 8.

**SegRWMA.** The SegRWMA dataset includes 198 patients with regional wall motion annotations, comprising 14 abnormal cases in the A4C view, 13 in the A3C view, and 12 in the A2C view, with the remaining patients considered normal. Segmentation masks are provided for the annotated frames, and we use the first annotated frame index for evaluation. In this study, we restrict analysis to the 2D ultrasound modality, as it is more cost-effective than contrast-enhanced echocardiography Liu et al. (2023). To prevent data leakage, the dataset is split at the patient level, ensuring that no patient appears in multiple splits. As shown in Figure 10, the abnormality distribution is imbalanced across splits: in the A2C view, 4 abnormal patients are in training, 5 in testing, and 3 in validation; in the A3C view, 6 are in training, 4 in testing, and 3 in validation; and in the A4C view, 6 are in training, 4 in testing, and 4 in validation. The remaining patients in each split are normal.

**CardiacNet.** The CardiacNet dataset contains 228 videos for ASD and 529 videos for PAH. Following the authors Yang et al. (2024), we treat each video as a separate patient. The dataset is

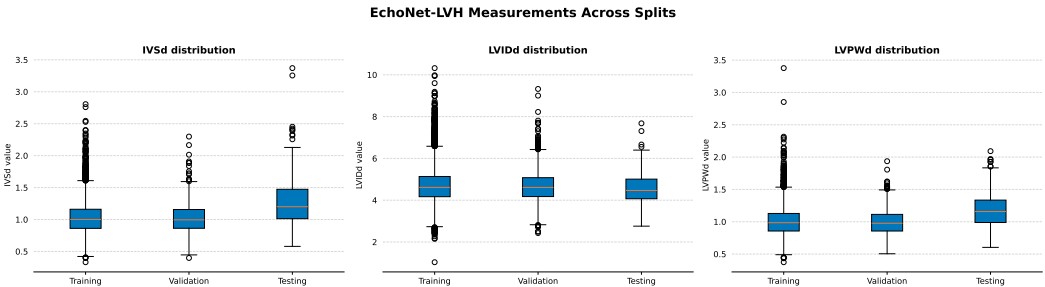

Figure 9: Distribution of structural measurements in the EchoNet-LVH dataset.

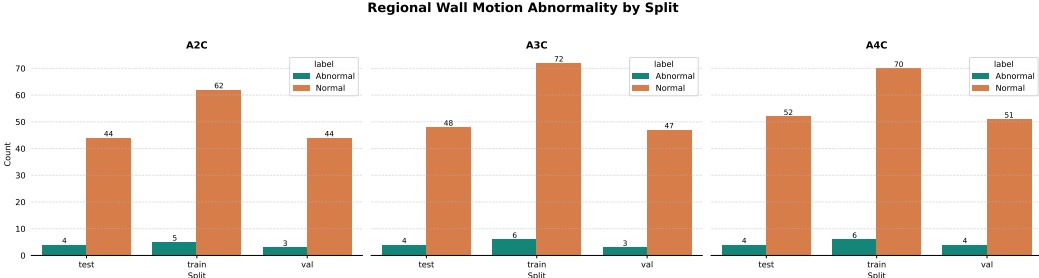

Figure 10: Distribution of regional wall motion abnormalities in the SegRWMA dataset across A2C, A3C, and A4C views and dataset splits.

divided independently for each task according to its distribution. For the CardiacNet-ASD subset, we apply a stratified split to preserve the proportion of ASD and non-ASD cases across subsets: 20% of patients are held out for testing, while the remaining 80% are further split, with 12.5% allocated to validation. For the CardiacNet-PAH subset, we use patient-level labels and again perform a stratified split to preserve the proportion of PAH and non-PAH cases: 20% of patients are reserved for testing, and from the remaining 80%, 12.5% are allocated to validation. The distribution of binary labels across splits for both ASD and PAH tasks is shown in Figure 11.

**HMC-QU.** The HMC-QU dataset contains 332 videos of A4C and A2C views with STEMI labels. Using patient-level labels, we apply a stratified split to maintain the STEMI/non-STEMI ratio across subsets. The dataset is divided into approximately 70.8% for training, 14.2% for validation, and 15% for testing, ensuring that all videos from the same patient remain in a single subset. We treat each video as a separate test case due to the relatively small dataset size. The distribution of STEMI and non-STEMI cases across splits is shown in Figure 12a.

**TMED-2.** TMED-2 is the only image dataset in our study, comprising 17,270 images across views: 1,670 A2C, 2,206 A4C, 4,808 PLAX, 1,725 PSAX, and 6,861 labeled as Other (A2C, A4C, or other views). Since many images belong to the same study, they are grouped into 598 studies in total. Following the official DEV479 split, the dataset is partitioned into 360 studies for training, 119 for validation, and 119 for testing. We also binarize the labels from multiclass classification into aortic stenosis "present" and "absent." The distribution of binary aortic stenosis labels across splits is presented in Figure 12b.

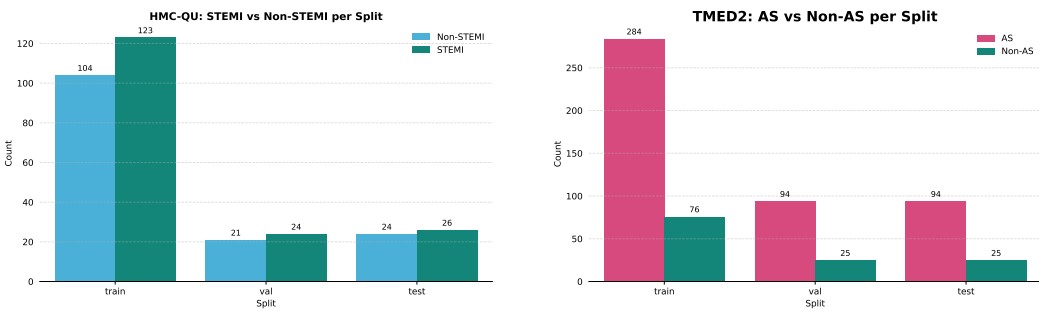

Figure 11: Distribution of binary labels in the CardiacNet dataset for ASD and PAH across training, validation, and test splits.

(a) Distribution of STEMI and non-STEMI cases in the HMC-QU dataset across training, validation, and test splits.

(b) Distribution of aortic stenosis cases (present vs. absent) in the TMED-2 dataset across training, validation, and test splits.

Figure 12: Overview of label distributions across splits for the HMC-QU and TMED-2 datasets.

# D    REPRODUCIBILITY

Each ultrasound video $V \in \Re^{T \times H \times W}$ is represented by 16 consecutive frames, normalized and resized to $224 \times 224$, yielding $X \in \Re^{16 \times 224 \times 224}$. A video encoder $f_\theta$ produces an embedding $z_v = f_\theta(X) \in \Re^d$, while a text prompt $P$ is mapped by a text encoder $g_\theta$ into $z_p = g_\theta(P) \in \Re^d$. For models originally designed for single images, we extend them to videos by computing predictions frame-wise and reporting the mean of the outputs across the 16 frames.

**Zero-shot evaluation.** For classification, we define one prompt per class $(P_1, \ldots, P_k)$ and predict using cosine similarity: $\hat{y} = \arg\max_c \cos(z_v, z_{p_c})$. This $\arg\max$ rule avoids dataset-specific thresholds, ensuring a calibration-free and reproducible evaluation. For regression tasks, we follow Christensen et al. (2023) by constructing prompts with numerical values over a predefined range. Predictions are obtained by aggregating frame-wise similarities (median of the top 20% per frame, averaged across frames). Prompt templates and robustness checks are detailed in Appendix E.

**Probing.** We assess the quality of the learned representations applying two lightweight classifiers directly on the embedding space. First, we perform linear probing by freezing the model's parameters and training a linear classifier on top of the embeddings. Linear probing tests whether the information needed for a task is linearly accessible. Second, for view classification task, we apply $k$-nearest neighbor (kNN) classification directly in the embedding space. Unlike linear probing, kNN evaluates whether local structure in the embedding space naturally reflects clinically meaningful view categories. By combining linear probing for global linear separability with kNN for local structure, we obtain complementary insights into how foundation models encode clinical information.

Training is carried out using the AdamW optimizer on the linear head only using a learning rate of 1e-4 with a weight decay of 1e-2. We use a batch size of 64, applying cross-entropy loss for classification tasks and mean squared error (MSE) loss for regression tasks. Early stopping is applied on the validation split to prevent overfitting. All experiments are conducted on an NVIDIA RTX A6000 GPU.

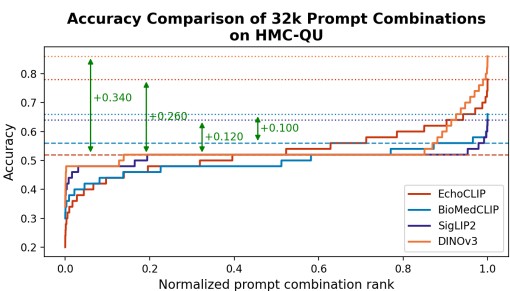

Figure 13: Accuracy over 32k prompt combinations on HMC-QU STEMI classification. Solid curves show accuracy trends, dotted lines indicate peak accuracies, and dashed lines mark baseline accuracies.

## E  PROMPTS

The prompt design follows the standard established by Christensen et al. (2023). Their exact ejection fraction prompt is used directly, while the prompts for the remaining tasks are generated in accordance with the same style. To improve robustness and reduce prompt-specific bias, we instantiate multiple phrasings per class (classification) or per numeric value (regression). For classification, the mean similarity is computed separately for each class and the class with the higher mean is selected. For regression, numerical placeholders are replaced with candidate values from a predefined grid, and the value corresponding to the prompt with the highest similarity is selected as the prediction. Specifically, ejection fraction is instantiated over integer values from 0–100%, while chamber dimensions and wall thicknesses are instantiated over clinically reasonable ranges with 0.1 cm resolution: LVIDd from 2.0–8.0 cm, IVSd from 0.5–2.0 cm, and LVPWd from 0.5–2.0 cm. All prompts and instantiation ranges are released on GitHub to ensure reproducibility.

To examine the effect of prompt design on model performance, we experimented with 32k different prompt combinations on the HMC-QU classification dataset. Our generated prompts are fixed as the baseline, and we additionally create variants with both relevant and irrelevant details. Figure 13 illustrates the gain in accuracy achieved by alternative combinations. Accuracy remains largely unchanged across most combinations, though a subset yields noticeable improvements. Importantly, the models do not reach their highest accuracy on the same prompt combination. Therefore, we retain our original prompts as the baseline choice.

Example of a regression-based prompt where the numerical variable is changed to a number with a predefined range.

```
"ejection_fraction":
  "THE LEFT VENTRICULAR EJECTION FRACTION IS ESTIMATED TO BE <#>%",
  "LV EJECTION FRACTION IS <#>%."
"LVIDd":
  "LEFT VENTRICULAR INTERNAL DIAMETER IN DIASTOLE (LVIDD) IS <#> CM.",
  "LVIDD IS <#> CM.",
```

Example of classification prompts

```
"Aortic Stenosis positive class":
  "AORTIC STENOSIS IS PRESENT. ",
  "SEVERE AORTIC STENOSIS. ",
  "CALCIFIED AORTIC VALVE WITH RESTRICTED LEAFLET MOTION. ",
"Aortic Stenosis negative class":
  "NO AORTIC STENOSIS. ",
  "NO SIGNIFICANT AORTIC VALVE STENOSIS. ",
  "AORTIC VALVE OPENS NORMALLY WITHOUT STENOSIS. ",
```

# F  ADDITIONAL EVALUATION

**Representation alignment across models.**   We study how encoders structure echocardiogram videos by comparing their embedding spaces with Canonical Correlation Analysis (CCA) and Centered Kernel Alignment (CKA). CCA provides low-dimensional projections for qualitative comparison, while CKA yields a scale- and rotation-invariant similarity score in $[0, 1]$. Figure 14 shows the two most correlated CCA dimensions when aligning DINOv3 to other models on CAMUS. The top row visualizes the distribution of DINOv3 embeddings colored by ejection-fraction (EF) bins; the bottom row shows the corresponding aligned coordinates for SigLIP2, BioMedCLIP, EchoCLIP, EchoPrime, and PanEcho, alongside their CKA similarity to DINOv3. Models with higher CKA tend to preserve DINOv3's large-scale geometry in the aligned space, suggesting more compatible feature organization.

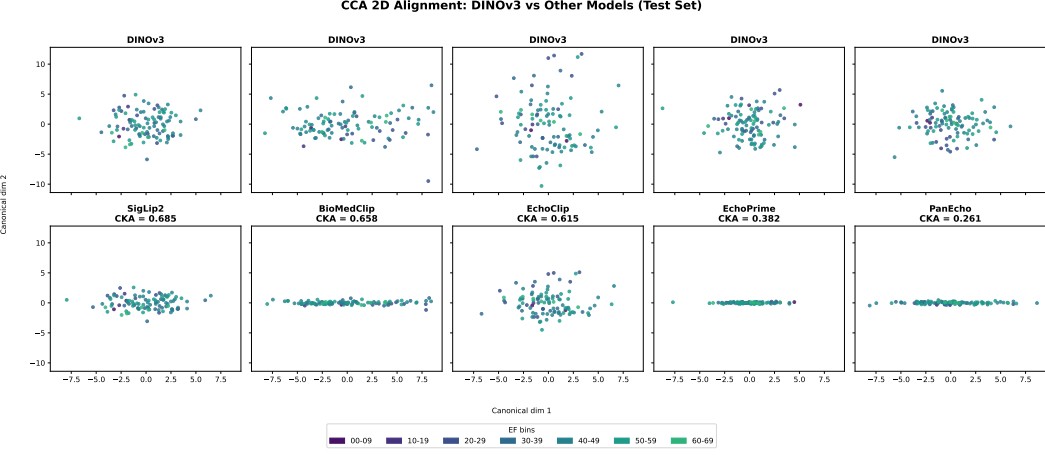

Figure 14: CCA 2D alignment between DINOv3 and other models on the CAMUS test set. Top row shows the distribution of DINOv3 embeddings across canonical dimensions, colored by EF bins. Bottom row shows aligned embeddings for SigLIP2, BioMedCLIP, EchoCLIP, EchoPrime, and PanEcho, with corresponding CKA similarity values to DINOv3.

**Relational structure via RSMs.**   To examine patient-level relationships, we compute Representation Similarity Matrices (RSMs) that capture pairwise cosine similarities within each model (Fig. 15). Difference maps against DINOv3 highlight where models agree (lighter) or diverge (darker) in inter-patient structure. On CAMUS, BioMedCLIP's RSM is more consistent with DINOv3 than EchoCLIP's, indicating closer relational alignment among generic vision-language and vision-only encoders than with echo-specialized ones.

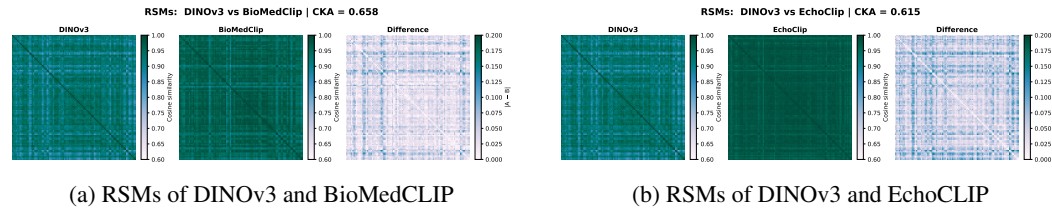

(a) RSMs of DINOv3 and BioMedCLIP                    (b) RSMs of DINOv3 and EchoCLIP

Figure 15: RSMs for DINOv3 and other models on the CAMUS test set. Matrices show pairwise cosine similarity between patient embeddings, while difference plots highlight agreement (light) or divergence (dark) in inter-patient relationships relative to DINOv3.

**Cross-dataset agreement.**   At the dataset level, correlations between model-wise embedding statistics are generally weak (Fig. 16, left). A PCA of the inter-model correlation matrix (Fig. 16, right) shows models scatter rather than forming a tight cluster, with only a mild tendency for general-purpose encoders to lie closer together. Overall, the models impose distinct relational geometries

on the same data, underscoring that comparable downstream scores can arise from meaningfully different internal organizations.

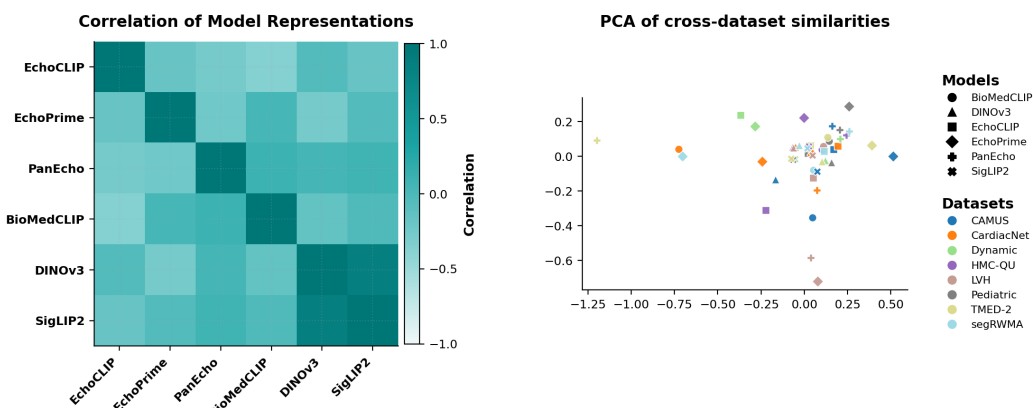

Figure 16: Correlation matrices of model embeddings (left) quantify how similarly models structure echocardiogram datasets, while PCA projections of these correlations (right) provide a 2D visualization that reveals groupings among general-purpose and echo-specific encoders

**Clustering consistency.** We further assess how well embeddings recover clinically meaningful view categories (A2C, A4C, PLAX, PSAX, Other) using the Adjusted Rand Index (ARI), which corrects for chance. Higher ARI indicates that a model's local neighborhoods align with true view labels, complementing CKA (global structure) and RSMs (pairwise relations) by probing cluster fidelity in the latent space (examples in Figs. 17 and 18).

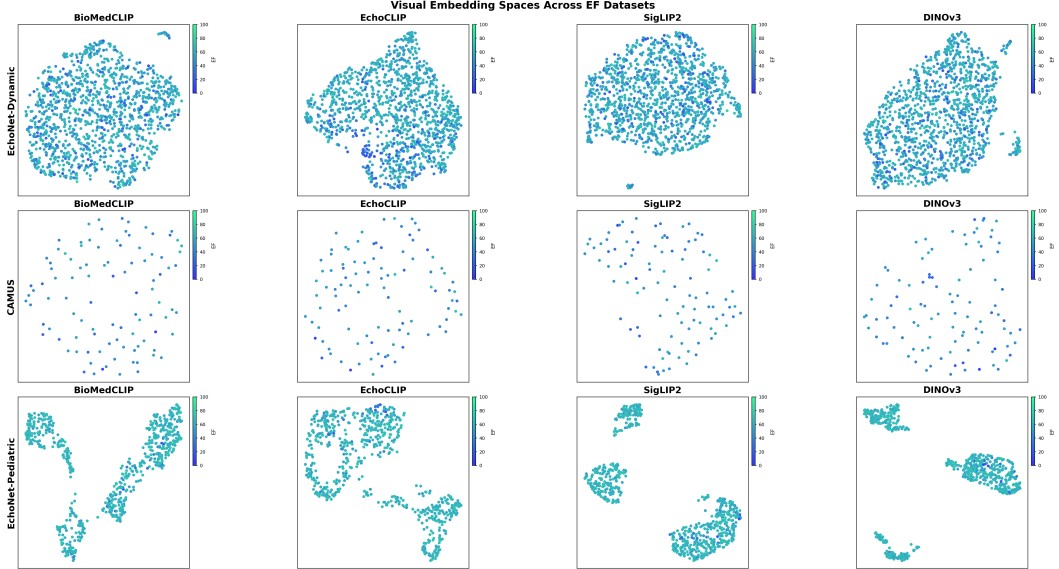

Figure 17: UMAP of visual representations on EchoNet-Dynamic, CAMUS, EchoNet-Pediatric Datasets

**Prompt–embedding alignment.** We visualize how visual and text embeddings interact across three clinical tasks (ASD, PAH, and STEMI) in Figs. 19–21. For each dataset, the top row shows 2D projections of visual embeddings colored by disease status, the middle row shows the corresponding positive and negative text prompt embeddings, and the bottom row depicts projection margins indicating alignment with the "Present" prompt. These visualizations highlight how well

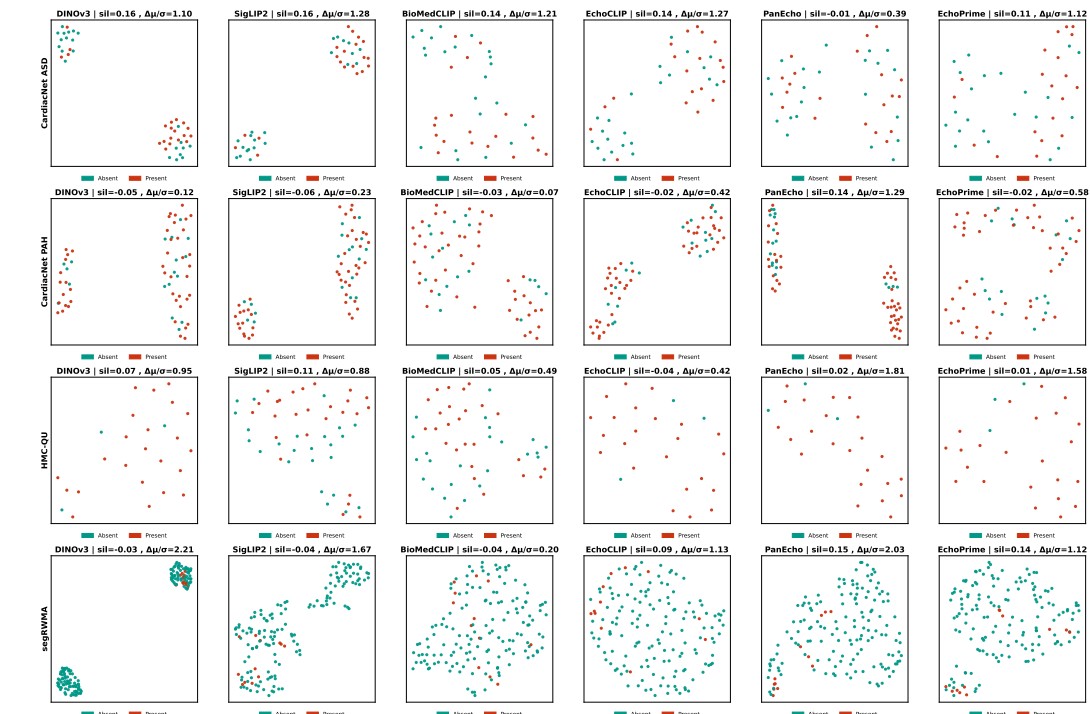

Figure 18: UMAP of visual representations on CardiacNet, HMC-QU, and segRWMA datasets.

foundation models separate disease classes in the latent space and reveal the degree to which visual embeddings align with textual supervision, offering qualitative insight that complements quantitative performance metrics. In the case of ASD (Fig. 19), DINOv3 achieves a clean separation of absent and present cases in the visual space, while SigLIP2 and BioMedCLIP show partial overlap between classes. However, the projection plots reveal that DINOv3 margins remain consistently negative with limited class separation, SigLIP2 exhibits small but coherent margins, BioMedCLIP shows weak alignment centered near zero, and EchoCLIP margins are widely dispersed across both groups.

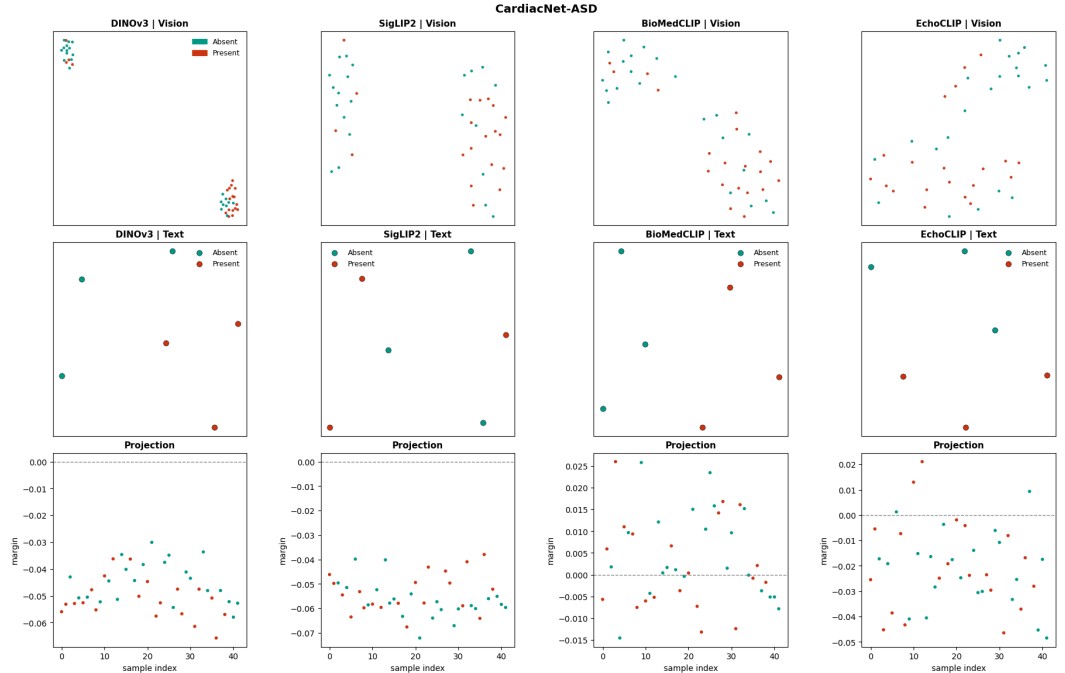

Figure 19: Embedding visualizations on the CardiacNet-ASD test set. Top: visual embeddings colored by ASD status. Middle: text prompt embeddings. Bottom: projection margins showing alignment with the "Present" prompt.

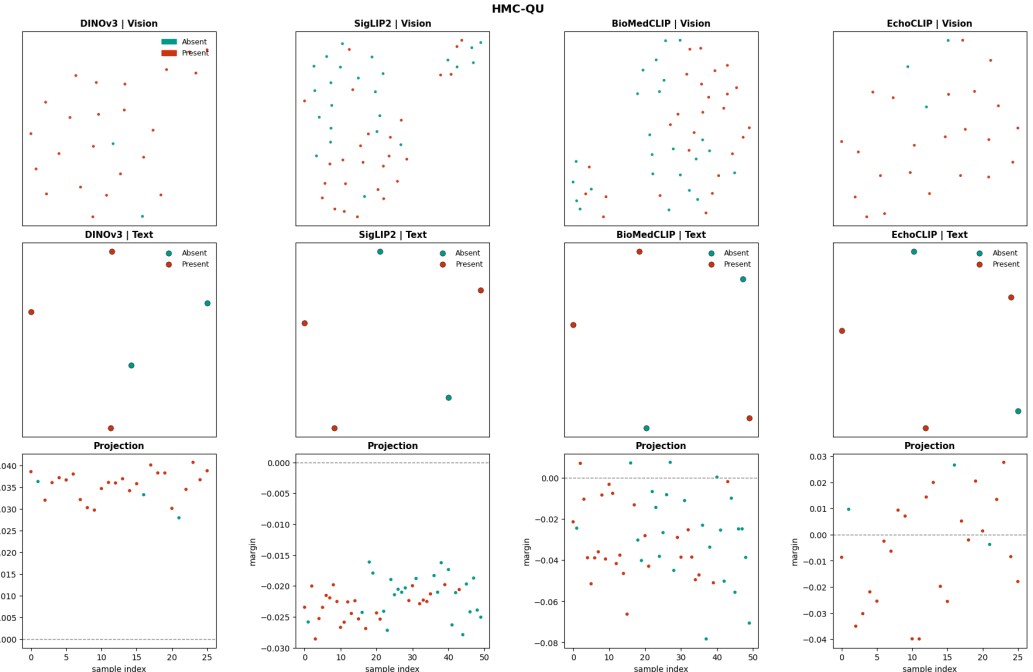

Figure 21: Embedding visualizations on the HMC-QU test set. Top: visual embeddings colored by STEMI status. Middle: text prompt embeddings. Bottom: projection margins showing alignment with the "Present" prompt.

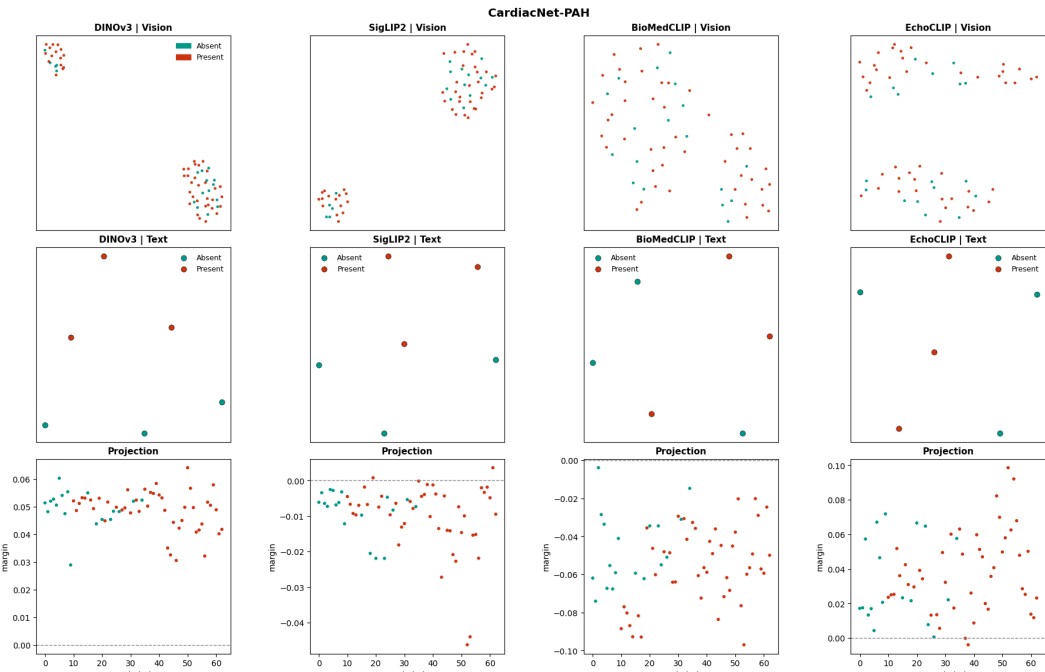

Figure 20: Embedding visualizations on the CardiacNet-PAH test set. Top: visual embeddings colored by PAH status. Middle: text prompt embeddings. Bottom: projection margins showing alignment with the "Present" prompt.

# G FULL RESULTS

Table 8: Zero-shot and linear probing performance on structural targets in EchoNet-LVH with 95% confidence intervals.

| Dataset | Model | Target | $n$ | MAE Raw | MAE Mean | MAE CI | NMAE (%) Raw | NMAE (%) Mean | NMAE (%) CI | RMSE Raw | RMSE Mean | RMSE CI | $R^2$ (%) Raw | $R^2$ (%) Mean | $R^2$ (%) CI |
|---|---|---|---|---|---|---|---|---|---|---|---|---|---|---|---|
| Zero | EchoCLIP | IVSd | 339 | 0.57 | 0.58 | [0.54–0.61] | 20.60 | 22.38 | [19.62–32.02] | 0.67 | 0.67 | [0.63–0.72] | -203.06 | -207.37 | [-262.66—157.47] |
| | PanEcho | IVSd | 339 | 0.21 | 0.21 | [0.19–0.23] | 7.45 | 8.10 | [6.86–11.63] | 0.30 | 0.30 | [0.26–0.34] | 39.67 | 39.27 | [32.14–46.50] |
| | BioMedCLIP | IVSd | 339 | 0.28 | 0.28 | [0.25–0.32] | 10.18 | 11.07 | [9.45–15.81] | 0.41 | 0.41 | [0.36–0.47] | -13.80 | -14.34 | [-23.33—6.07] |
| | DINOv3 | IVSd | 339 | 0.28 | 0.28 | [0.26–0.31] | 10.16 | 11.02 | [9.52–15.54] | 0.39 | 0.39 | [0.34–0.44] | -0.36 | -0.66 | [-2.54–0.00] |
| | SigLIP2 | IVSd | 339 | 0.28 | 0.28 | [0.26–0.31] | 10.15 | 11.01 | [9.50–15.52] | 0.39 | 0.39 | [0.34–0.44] | -0.42 | -0.71 | [-2.67–0.00] |
| | EchoCLIP | LVIDd | 340 | 0.79 | 0.79 | [0.73–0.85] | 16.12 | 17.28 | [15.19–21.61] | 0.97 | 0.97 | [0.90–1.03] | -87.10 | -89.38 | [-125.56—60.11] |
| | PanEcho | LVIDd | 340 | 0.36 | 0.36 | [0.33–0.39] | 7.28 | 7.80 | [6.87–9.68] | 0.45 | 0.45 | [0.41–0.49] | 59.90 | 59.55 | [52.02–66.04] |
| | BioMedCLIP | LVIDd | 340 | 0.97 | 0.97 | [0.90–1.03] | 19.67 | 21.09 | [18.68–26.80] | 1.14 | 1.14 | [1.07–1.21] | -160.88 | -164.43 | [-221.77—117.61] |
| | DINOv3 | LVIDd | 340 | 0.69 | 0.69 | [0.64–0.74] | 13.98 | 14.99 | [13.26–18.76] | 0.84 | 0.83 | [0.78–0.89] | -39.08 | -40.39 | [-60.89—25.07] |
| | SigLIP2 | LVIDd | 340 | 0.69 | 0.69 | [0.64–0.74] | 13.99 | 15.00 | [13.27–18.78] | 0.84 | 0.83 | [0.78–0.89] | -39.23 | -40.54 | [-61.09—25.17] |
| | EchoCLIP | LVPWd | 340 | 0.41 | 0.41 | [0.38–0.44] | 27.63 | 29.08 | [25.96–32.93] | 0.50 | 0.50 | [0.47–0.53] | -249.10 | -251.58 | [-300.51—211.68] |
| | PanEcho | LVPWd | 340 | 0.18 | 0.18 | [0.17–0.19] | 12.04 | 12.68 | [11.35–14.34] | 0.23 | 0.23 | [0.21–0.24] | 28.59 | 28.13 | [16.79–37.97] |
| | BioMedCLIP | LVPWd | 340 | 0.26 | 0.26 | [0.24–0.28] | 17.59 | 18.49 | [16.57–20.86] | 0.32 | 0.32 | [0.30–0.34] | -42.39 | -43.16 | [-62.80—28.40] |
| | DINOv3 | LVPWd | 340 | 0.22 | 0.22 | [0.21–0.24] | 15.02 | 15.79 | [14.13–17.81] | 0.28 | 0.28 | [0.26–0.30] | -6.73 | -7.11 | [-15.05—2.24] |
| | SigLIP2 | LVPWd | 340 | 0.22 | 0.22 | [0.21–0.24] | 15.03 | 15.80 | [14.14–17.82] | 0.28 | 0.28 | [0.26–0.30] | -6.83 | -7.23 | [-15.22—2.30] |
| Linear Probing | EchoCLIP | IVSd | 339 | 0.28 | 0.29 | [0.25–0.32] | 10.20 | 11.10 | [9.36–15.83] | 0.42 | 0.42 | [0.37–0.48] | -20.16 | -21.03 | [-32.48—11.33] |
| | EchoPrime | IVSd | 339 | 0.25 | 0.25 | [0.22–0.28] | 9.00 | 9.79 | [8.24–13.90] | 0.38 | 0.38 | [0.33–0.43] | 3.55 | 3.05 | [-6.04–11.64] |
| | PanEcho | IVSd | 339 | 0.18 | 0.18 | [0.16–0.20] | 6.47 | 7.03 | [6.00–9.98] | 0.27 | 0.26 | [0.23–0.31] | 52.76 | 52.51 | [46.54–58.44] |
| | BioMedCLIP | IVSd | 339 | 0.30 | 0.30 | [0.27–0.33] | 10.61 | 11.54 | [9.83–16.49] | 0.42 | 0.42 | [0.37–0.47] | -18.36 | -19.19 | [-30.36—9.25] |
| | DINOv3 | IVSd | 339 | 0.27 | 0.28 | [0.25–0.31] | 9.84 | 10.71 | [9.04–15.30] | 0.40 | 0.40 | [0.35–0.46] | -8.93 | -9.79 | [-19.98—0.94] |
| | SigLIP2 | IVSd | 339 | 0.30 | 0.30 | [0.27–0.34] | 10.86 | 11.82 | [10.02–16.83] | 0.44 | 0.44 | [0.38–0.49] | -28.17 | -29.15 | [-41.49—18.49] |
| | EchoFM | IVSd | 339 | 0.32 | 0.32 | [0.29–0.36] | 11.59 | 12.63 | [10.75–17.93] | 0.46 | 0.46 | [0.40–0.52] | -41.10 | -42.28 | [-55.21—31.12] |
| | EchoCLIP | LVIDd | 340 | 0.47 | 0.47 | [0.43–0.51] | 9.55 | 10.23 | [8.93–12.71] | 0.60 | 0.59 | [0.54–0.65] | 29.30 | 28.72 | [16.40–39.24] |
| | EchoPrime | LVIDd | 340 | 0.41 | 0.41 | [0.37–0.44] | 8.30 | 8.90 | [7.70–11.06] | 0.54 | 0.54 | [0.48–0.59] | 42.44 | 41.83 | [28.67–53.52] |
| | PanEcho | LVIDd | 340 | 0.35 | 0.35 | [0.32–0.38] | 7.16 | 7.67 | [6.64–9.64] | 0.45 | 0.45 | [0.41–0.49] | 58.87 | 58.41 | [48.50–67.55] |
| | BioMedCLIP | LVIDd | 340 | 0.52 | 0.52 | [0.47–0.56] | 10.51 | 11.26 | [9.89–14.01] | 0.65 | 0.65 | [0.60–0.71] | 15.14 | 14.51 | [5.72–21.56] |
| | DINOv3 | LVIDd | 340 | 0.47 | 0.47 | [0.43–0.51] | 9.55 | 10.22 | [8.96–12.54] | 0.60 | 0.60 | [0.55–0.66] | 28.14 | 27.82 | [19.30–35.79] |
| | SigLIP2 | LVIDd | 340 | 0.51 | 0.51 | [0.47–0.56] | 10.38 | 11.12 | [9.75–13.74] | 0.65 | 0.65 | [0.59–0.71] | 15.34 | 14.86 | [5.85–22.72] |
| | EchoFM | LVIDd | 340 | 0.57 | 0.57 | [0.52–0.62] | 11.59 | 12.41 | [10.97–15.37] | 0.71 | 0.71 | [0.66–0.78] | -1.85 | -2.37 | [-8.57–2.29] |
| | EchoCLIP | LVPWd | 340 | 0.22 | 0.22 | [0.20–0.24] | 14.59 | 15.36 | [13.53–17.63] | 0.29 | 0.29 | [0.26–0.32] | -17.64 | -18.24 | [-30.21—6.74] |
| | EchoPrime | LVPWd | 340 | 0.19 | 0.19 | [0.17–0.21] | 12.73 | 13.41 | [11.91–15.31] | 0.25 | 0.25 | [0.23–0.28] | 9.10 | 8.65 | [-2.06–18.58] |
| | PanEcho | LVPWd | 340 | 0.15 | 0.15 | [0.14–0.16] | 9.96 | 10.48 | [9.41–11.87] | 0.19 | 0.19 | [0.17–0.21] | 48.89 | 48.58 | [40.67–55.93] |
| | BioMedCLIP | LVPWd | 340 | 0.23 | 0.23 | [0.21–0.25] | 15.18 | 15.98 | [14.11–18.17] | 0.30 | 0.30 | [0.27–0.33] | -25.09 | -25.68 | [-38.30—14.00] |
| | DINOv3 | LVPWd | 340 | 0.21 | 0.21 | [0.19–0.23] | 14.11 | 14.86 | [13.21–16.93] | 0.28 | 0.28 | [0.25–0.30] | -8.57 | -9.11 | [-20.28—1.02] |
| | SigLIP2 | LVPWd | 340 | 0.23 | 0.23 | [0.21–0.25] | 15.15 | 15.95 | [14.08–18.20] | 0.30 | 0.30 | [0.27–0.33] | -27.05 | -27.68 | [-40.50—16.37] |
| | EchoFM | LVPWd | 340 | 0.24 | 0.24 | [0.22–0.27] | 16.30 | 17.16 | [15.19–19.56] | 0.32 | 0.32 | [0.29–0.35] | -43.83 | -44.57 | [-58.40—30.70] |

Table 7: EF zero-shot results on EchoNet-Dynamic, CAMUS, and EchoNet-Pediatric with 95% confidence intervals. The *View* column indicates the view of the ground truth: EchoNet-Dynamic corresponds to A4C; CAMUS includes both A2C and A4C. In the main paper, results are reported at the study level, but here we additionally provide view-specific results. For EchoNet-Pediatric, we report results for each individual view as well as for the combined set of views.

| Dataset | Model | View | n | MAE Raw | Mean | CI | NMAE (%) Raw | Mean | CI | RMSE Raw | Mean | CI | $R^2$ (%) Raw | Mean | CI |
|---|---|---|---|---|---|---|---|---|---|---|---|---|---|---|---|
| EchoNet-Dynamic | EchoCLIP | A4C | 1277 | 9.99 | 9.98 | [9.53–10.42] | 13.45 | 13.79 | [12.96–15.01] | 12.99 | 12.98 | [12.38–13.55] | -12.94 | -12.91 | [-25.60—-0.89] |
| | EchoPrime | A4C | 1277 | 7.78 | 7.79 | [7.35–8.20] | 10.48 | 10.76 | [10.04–11.79] | 10.87 | 10.87 | [10.29–11.42] | 21.00 | 20.93 | [16.17–25.40] |
| | PanEcho | A4C | 1277 | 5.79 | 5.79 | [5.50–6.10] | 7.79 | 8.00 | [7.45–8.76] | 8.11 | 8.11 | [7.65–8.59] | 56.00 | 55.93 | [50.11–61.27] |
| | BioMedCLIP | A4C | 1277 | 13.83 | 13.83 | [13.45–14.25] | 18.63 | 19.11 | [18.26–20.85] | 15.62 | 15.62 | [15.22–16.04] | -63.23 | -63.49 | [-77.62—-51.07] |
| | DINOv3 | A4C | 1277 | 14.67 | 14.67 | [14.31–15.07] | 19.75 | 20.27 | [19.37–22.07] | 16.13 | 16.13 | [15.80–16.49] | -73.97 | -74.37 | [-91.10—-59.67] |
| | SigLIP2 | A4C | 1277 | 14.66 | 14.66 | [14.30–15.06] | 19.74 | 20.26 | [19.36–22.06] | 16.11 | 16.12 | [15.79–16.48] | -73.75 | -74.15 | [-90.84—-59.48] |
| | EchoPrime with Shuffled frames | A4C | 1277 | 6.12 | 6.12 | [5.79–6.43] | 8.24 | 8.46 | [7.91–9.24] | 8.36 | 8.36 | [7.90–8.81] | 53.18 | 53.14 | [48.50–57.77] |
| | PanEcho with Shuffled frames | A4C | 1277 | 6.31 | 6.32 | [6.02–6.61] | 8.50 | 8.73 | [8.17–9.51] | 8.49 | 8.49 | [8.11–8.89] | 51.81 | 51.65 | [45.15–57.28] |
| CAMUS | EchoCLIP | A2C | 50 | 9.85 | 9.82 | [7.87–12.02] | 15.39 | 17.88 | [13.46–24.74] | 12.25 | 12.14 | [9.83–14.71] | -0.41 | -4.72 | [-47.33–22.29] |
| | PanEcho | A2C | 50 | 11.85 | 11.78 | [9.50–14.16] | 18.51 | 21.56 | [15.82–30.96] | 14.59 | 14.45 | [11.80–17.03] | -42.32 | -51.66 | [-143.87–8.91] |
| | BioMedCLIP | A2C | 50 | 17.88 | 17.85 | [14.94–21.43] | 27.94 | 32.54 | [25.02–44.12] | 21.35 | 21.18 | [17.52–25.68] | -204.71 | -218.79 | [-340.13—-134.36] |
| | DINOv3 | A2C | 50 | 9.88 | 9.85 | [7.56–12.49] | 15.43 | 17.87 | [13.34–23.88] | 13.32 | 13.14 | [10.04–16.78] | -18.54 | -20.84 | [-51.18—-3.18] |
| | SigLIP2 | A2C | 50 | 9.28 | 9.25 | [7.18–11.54] | 14.50 | 16.79 | [12.51–22.29] | 12.23 | 12.09 | [9.44–15.17] | -0.06 | -2.15 | [-11.20—-0.01] |
| | EchoCLIP | A4C | 50 | 10.41 | 10.41 | [8.18–12.77] | 16.27 | 18.93 | [14.11–24.90] | 13.35 | 13.25 | [10.09–16.45] | -19.20 | -23.97 | [-66.46–7.91] |
| | PanEcho | A4C | 50 | 11.95 | 11.94 | [9.52–14.46] | 18.67 | 21.86 | [15.68–30.93] | 14.99 | 14.91 | [12.03–17.78] | -50.23 | -61.26 | [-156.34–3.19] |
| | BioMedCLIP | A4C | 50 | 19.87 | 19.79 | [16.53–23.59] | 31.05 | 36.09 | [28.02–49.23] | 23.55 | 23.32 | [19.10–28.11] | -270.95 | -287.51 | [-442.03—-178.65] |
| | DINOv3 | A4C | 50 | 9.88 | 9.85 | [7.56–12.49] | 15.43 | 17.87 | [13.34–23.88] | 13.32 | 13.14 | [10.04–16.78] | -18.55 | -20.85 | [-51.19—-3.19] |
| | SigLIP2 | A4C | 50 | 9.28 | 9.25 | [7.18–11.54] | 14.50 | 16.79 | [12.51–22.29] | 12.23 | 12.09 | [9.43–15.17] | -0.06 | -2.16 | [-11.20—-0.01] |
| | EchoCLIP | study-level | 50 | 9.83 | 9.82 | [7.89–11.95] | 15.36 | 17.86 | [13.40–23.84] | 12.31 | 12.20 | [9.51–14.84] | -1.25 | -5.04 | [-36.53–19.61] |
| | EchoPrime | study-level | 50 | 14.00 | 13.92 | [11.33–17.00] | 21.88 | 25.18 | [19.14–34.04] | 17.37 | 17.16 | [13.64–21.28] | -101.92 | -110.04 | [-199.12—-43.96] |
| | PanEcho | study-level | 50 | 11.63 | 11.59 | [9.28–14.05] | 18.17 | 21.22 | [15.35–29.81] | 14.53 | 14.43 | [11.74–17.07] | -41.13 | -50.91 | [-143.25–9.65] |
| | BioMedCLIP | study-level | 50 | 18.87 | 18.81 | [15.71–22.48] | 29.48 | 34.30 | [26.61–46.82] | 22.33 | 22.13 | [18.21–26.79] | -233.33 | -248.29 | [-387.24—-154.42] |
| | DINOv3 | study-level | 50 | 9.88 | 9.85 | [7.56–12.49] | 15.43 | 17.87 | [13.34–23.88] | 13.32 | 13.14 | [10.04–16.78] | -18.55 | -20.84 | [-51.18—-3.18] |
| | SigLIP2 | study-level | 50 | 9.28 | 9.25 | [7.18–11.54] | 14.50 | 16.79 | [12.51–22.29] | 12.23 | 12.09 | [9.43–15.17] | -0.06 | -2.16 | [-11.20—-0.01] |
| EchoNet-Pediatric | EchoCLIP | A4C | 271 | 10.15 | 10.14 | [9.10–11.16] | 16.77 | 17.92 | [15.26–21.72] | 13.01 | 12.97 | [11.75–14.13] | -125.55 | -137.02 | [-268.96—-51.76] |
| | EchoPrime | A4C | 271 | 5.59 | 5.59 | [5.01–6.17] | 9.23 | 9.86 | [8.49–11.79] | 7.54 | 7.52 | [6.59–8.53] | 24.16 | 22.12 | [3.85–34.60] |
| | PanEcho | A4C | 271 | 9.17 | 9.15 | [8.18–10.18] | 15.15 | 16.16 | [13.66–19.92] | 12.37 | 12.32 | [11.01–13.64] | -104.11 | -114.10 | [-235.28—-32.23] |
| | BioMedCLIP | A4C | 271 | 16.10 | 16.10 | [15.37–16.92] | 26.59 | 28.44 | [25.56–33.18] | 17.40 | 17.39 | [16.70–18.15] | -303.44 | -324.98 | [-531.45—-185.29] |
| | DINOv3 | A4C | 271 | 18.21 | 18.21 | [17.58–18.84] | 30.08 | 32.18 | [29.28–37.28] | 19.03 | 19.03 | [18.45–19.59] | -382.85 | -408.85 | [-648.17—-242.90] |
| | SigLIP2 | A4C | 271 | 18.19 | 18.19 | [17.56–18.82] | 30.05 | 32.14 | [29.25–37.24] | 19.01 | 19.01 | [18.43–19.57] | -381.92 | -407.85 | [-646.61—-242.28] |
| | EchoCLIP | PSAX | 387 | 16.35 | 16.35 | [15.24–17.47] | 26.93 | 28.66 | [25.41–33.30] | 19.55 | 19.54 | [18.46–20.62] | -478.81 | -489.65 | [-729.94—-311.59] |
| | EchoPrime | PSAX | 387 | 5.34 | 5.35 | [4.87–5.89] | 8.79 | 9.38 | [8.18–10.91] | 7.42 | 7.43 | [6.47–8.50] | 16.66 | 16.28 | [1.29–27.86] |
| | PanEcho | PSAX | 387 | 9.06 | 9.05 | [8.29–9.85] | 14.91 | 15.86 | [13.80–18.96] | 12.11 | 12.09 | [11.02–13.17] | -121.99 | -125.84 | [-223.92—-55.67] |
| | BioMedCLIP | PSAX | 387 | 19.84 | 19.82 | [19.19–20.41] | 32.67 | 34.74 | [31.78–39.60] | 20.80 | 20.78 | [20.22–21.34] | -555.14 | -567.34 | [-833.08—-368.31] |
| | DINOv3 | PSAX | 387 | 18.26 | 18.25 | [17.71–18.77] | 30.07 | 31.98 | [29.38–36.48] | 19.08 | 19.07 | [18.60–19.54] | -451.19 | -461.42 | [-688.79—-294.46] |
| | SigLIP2 | PSAX | 387 | 18.24 | 18.23 | [17.69–18.75] | 30.04 | 31.95 | [29.35–36.45] | 19.06 | 19.05 | [18.58–19.53] | -450.21 | -460.42 | [-687.34—-293.84] |
| | EchoCLIP | combined | 658 | 13.80 | 13.79 | [13.02–14.63] | 22.72 | 23.23 | [21.51–27.02] | 17.16 | 17.15 | [16.41–18.01] | -322.17 | -331.41 | [-461.64—-229.56] |
| | EchoPrime | combined | 658 | 5.44 | 5.44 | [5.07–5.84] | 8.96 | 9.17 | [8.40–10.50] | 7.47 | 7.45 | [6.78–8.22] | 20.01 | 19.30 | [8.61–27.40] |
| | PanEcho | combined | 658 | 9.10 | 9.10 | [8.47–9.68] | 14.99 | 15.33 | [14.02–17.99] | 12.22 | 12.19 | [11.35–13.01] | -113.99 | -118.30 | [-188.76—-62.86] |
| | BioMedCLIP | combined | 658 | 18.30 | 18.30 | [17.81–18.82] | 30.13 | 30.83 | [29.42–35.31] | 19.47 | 19.47 | [19.03–19.95] | -443.45 | -455.89 | [-625.26—-333.69] |
| | DINOv3 | combined | 658 | 18.24 | 18.24 | [17.84–18.65] | 30.03 | 30.73 | [29.41–35.09] | 19.06 | 19.06 | [18.71–19.42] | -420.73 | -432.63 | [-596.67—-316.12] |
| | SigLIP2 | combined | 658 | 18.22 | 18.22 | [17.82–18.64] | 30.01 | 30.70 | [29.38–35.05] | 19.04 | 19.04 | [18.69–19.41] | -419.78 | -431.65 | [-595.35—-315.37] |

Table 9: Zero-shot and linear probing performance on CardiacNet-PAH and CardiacNet-ASD tasks with 95% confidence intervals.

| Setting | Task | Model | n | Accuracy (%) Raw | Mean | CI | Balanced Accuracy (%) Raw | Mean | CI | F1 (%) Raw | Mean | CI |
|---|---|---|---|---|---|---|---|---|---|---|---|---|
| Zero-shot | PAH | EchoCLIP | 106 | 70.75 | 70.69 | [62.26–79.25] | 51.89 | 51.78 | [47.37–57.07] | 46.96 | 46.63 | [39.35–55.47] |
| | | BioMedCLIP | 106 | 29.25 | 29.34 | [20.75–37.74] | 46.22 | 46.39 | [39.74–52.27] | 25.75 | 25.66 | [18.94–33.07] |
| | | DINOv3 | 106 | 70.75 | 70.77 | [62.26–79.25] | 50.00 | 50.00 | [50.00–50.00] | 41.44 | 41.40 | [38.37–44.21] |
| | | SigLIP2 | 106 | 30.19 | 30.14 | [22.62–38.68] | 50.67 | 50.64 | [50.00–52.11] | 24.11 | 23.94 | [18.46–29.62] |
| | ASD | EchoCLIP | 47 | 53.19 | 53.04 | [38.30–68.09] | 59.26 | 59.16 | [52.08–67.86] | 47.88 | 47.13 | [33.55–62.78] |
| | | BioMedCLIP | 47 | 51.06 | 51.35 | [38.30–65.96] | 45.74 | 45.96 | [37.00–55.31] | 40.24 | 40.02 | [29.85–52.16] |
| | | DINOv3 | 47 | 57.45 | 57.47 | [42.55–72.34] | 50.00 | 50.00 | [50.00–50.00] | 36.49 | 36.35 | [29.85–41.98] |
| | | SigLIP2 | 47 | 57.45 | 57.47 | [42.55–72.34] | 50.00 | 50.00 | [50.00–50.00] | 36.49 | 36.35 | [29.85–41.98] |
| Linear Probing | PAH | EchoCLIP | 106 | 70.75 | 70.77 | [62.26–79.25] | 50.00 | 50.00 | [50.00–50.00] | 41.44 | 41.40 | [38.37–44.21] |
| | | EchoPrime | 106 | 73.58 | 73.45 | [64.15–82.08] | 62.41 | 62.38 | [53.34–72.44] | 63.36 | 63.02 | [52.23–74.30] |
| | | PanEcho | 106 | 72.64 | 72.77 | [64.15–82.08] | 60.80 | 61.00 | [51.62–70.45] | 61.51 | 61.45 | [50.30–72.29] |
| | | BioMedCLIP | 106 | 70.75 | 70.77 | [62.26–79.25] | 50.00 | 50.00 | [50.00–50.00] | 41.44 | 41.40 | [38.37–44.21] |
| | | EchoFM | 106 | 70.75 | 70.77 | [62.26–79.25] | 50.00 | 50.00 | [50.00–50.00] | 41.44 | 41.40 | [38.37–44.21] |
| | | DINOv3 | 106 | 73.58 | 73.60 | [65.09–81.13] | 59.57 | 59.60 | [51.14–68.30] | 59.85 | 59.59 | [48.24–70.31] |
| | | SigLIP2 | 106 | 72.64 | 72.68 | [64.15–81.13] | 53.23 | 53.25 | [50.00–57.81] | 47.96 | 47.83 | [40.11–56.56] |
| | ASD | EchoCLIP | 47 | 57.45 | 57.47 | [42.55–72.34] | 50.00 | 50.00 | [50.00–50.00] | 36.49 | 36.35 | [29.85–41.98] |
| | | EchoPrime | 47 | 53.19 | 53.35 | [38.30–68.09] | 56.02 | 56.19 | [42.39–69.55] | 52.66 | 52.29 | [37.28–65.94] |
| | | PanEcho | 47 | 61.70 | 61.99 | [48.94–74.47] | 58.89 | 59.12 | [45.98–72.35] | 58.53 | 58.18 | [44.05–72.57] |
| | | EchoFM | 47 | 53.19 | 53.12 | [38.30–68.09] | 50.83 | 50.77 | [36.97–64.76] | 50.48 | 49.91 | [35.05–64.00] |
| | | BioMedCLIP | 47 | 61.70 | 61.81 | [48.88–74.47] | 58.89 | 59.02 | [46.10–72.03] | 58.53 | 58.15 | [44.29–71.88] |
| | | DINOv3 | 47 | 59.57 | 59.62 | [44.68–74.47] | 57.04 | 57.08 | [43.43–70.11] | 56.76 | 56.24 | [42.12–70.41] |
| | | SigLIP2 | 47 | 70.21 | 70.31 | [55.32–82.98] | 68.24 | 68.34 | [55.18–81.57] | 68.49 | 68.07 | [53.98–81.58] |

Table 10: Zero-shot and linear probing performance on HMC-QU dataset. The *View* column indicates the view of the ground truth: HMC-QU includes both A2C and A4C. In the main paper, results are reported at the combined level, but here we additionally provide view-specific results.

| Setting | Model | View | $n$ | Accuracy (%) | | | Balanced Accuracy (%) | | | F1 (%) | | |
| --- | --- | --- | --- | --- | --- | --- | --- | --- | --- | --- | --- | --- |
| | | | | Raw | Mean | CI | Raw | Mean | CI | Raw | Mean | CI |
| Zero-shot | EchoCLIP | A2C | 25 | 52.00 | 51.94 | [32.00–72.00] | 53.85 | 53.58 | [50.00–62.50] | 40.48 | 39.30 | [25.00–57.14] |
| | BioMedCLIP | A2C | 25 | 44.00 | 44.12 | [24.00–64.00] | 45.83 | 45.75 | [36.36–50.00] | 30.56 | 30.27 | [19.35–39.02] |
| | DINOv3 | A2C | 25 | 48.00 | 48.20 | [28.00–68.00] | 50.00 | 50.00 | [50.00–50.00] | 32.43 | 32.20 | [21.88–40.48] |
| | SigLIP2 | A2C | 25 | 52.00 | 51.80 | [32.00–72.00] | 50.00 | 50.00 | [50.00–50.00] | 34.21 | 33.83 | [24.24–41.86] |
| | EchoCLIP | A4C | 25 | 64.00 | 63.84 | [44.00–84.00] | 62.01 | 61.90 | [41.91–81.17] | 61.80 | 60.69 | [40.46–80.20] |
| | BioMedCLIP | A4C | 25 | 56.00 | 55.71 | [36.00–72.10] | 50.00 | 50.00 | [50.00–50.00] | 35.90 | 35.52 | [26.47–41.89] |
| | DINOv3 | A4C | 25 | 56.00 | 55.71 | [36.00–72.10] | 50.00 | 50.00 | [50.00–50.00] | 35.90 | 35.52 | [26.47–41.89] |
| | SigLIP2 | A4C | 25 | 44.00 | 44.29 | [27.90–64.00] | 50.00 | 50.00 | [50.00–50.00] | 30.56 | 30.38 | [21.81–39.02] |
| | EchoCLIP | combined | 50 | 58.00 | 57.94 | [42.00–72.00] | 56.73 | 56.59 | [45.00–67.37] | 52.51 | 51.91 | [37.48–66.09] |
| | BioMedCLIP | combined | 50 | 50.00 | 50.26 | [36.00–64.00] | 48.08 | 48.18 | [44.12–50.00] | 33.33 | 33.29 | [26.47–39.02] |
| | DINOv3 | combined | 50 | 52.00 | 52.15 | [38.00–66.00] | 50.00 | 50.00 | [50.00–50.00] | 34.21 | 34.13 | [27.54–39.76] |
| | SigLIP2 | combined | 50 | 48.00 | 47.85 | [34.00–62.00] | 50.00 | 50.00 | [50.00–50.00] | 32.43 | 32.20 | [25.37–38.27] |
| Linear Probing | EchoCLIP | A2C | 25 | 76.00 | 75.70 | [56.00–92.00] | 75.96 | 75.58 | [55.84–91.88] | 75.96 | 74.83 | [55.72–91.88] |
| | EchoPrime | A2C | 25 | 84.00 | 83.95 | [68.00–96.00] | 83.97 | 83.93 | [68.26–96.67] | 83.97 | 83.34 | [67.51–96.00] |
| | PanEcho | A2C | 25 | 68.00 | 67.66 | [48.00–84.00] | 67.63 | 67.28 | [48.40–84.75] | 67.53 | 66.25 | [47.66–83.77] |
| | BioMedCLIP | A2C | 25 | 56.00 | 55.96 | [36.00–76.00] | 57.05 | 56.87 | [38.78–73.27] | 53.31 | 52.07 | [30.56–72.46] |
| | DINOv3 | A2C | 25 | 80.00 | 79.81 | [64.00–96.00] | 80.77 | 80.47 | [66.64–94.44] | 79.47 | 78.44 | [60.32–95.54] |
| | SigLIP2 | A2C | 25 | 80.00 | 79.81 | [64.00–96.00] | 80.77 | 80.47 | [66.64–94.44] | 79.47 | 78.44 | [60.32–95.54] |
| | EchoFM | A2C | 25 | 68.00 | 67.60 | [48.00–84.00] | 67.63 | 67.22 | [48.33–84.29] | 67.53 | 66.16 | [46.63–83.77] |
| | EchoCLIP | A4C | 25 | 72.00 | 71.86 | [52.00–88.00] | 73.05 | 72.92 | [55.14–88.34] | 72.00 | 71.06 | [51.92–87.92] |
| | EchoPrime | A4C | 25 | 76.00 | 76.02 | [59.90–92.00] | 76.62 | 76.54 | [58.76–91.88] | 75.96 | 75.20 | [56.00–91.67] |
| | PanEcho | A4C | 25 | 72.00 | 72.10 | [56.00–88.00] | 74.03 | 74.05 | [57.01–88.90] | 71.82 | 71.05 | [51.91–87.92] |
| | EchoFM | A4C | 25 | 76.00 | 75.84 | [60.00–92.00] | 77.60 | 77.33 | [62.15–91.67] | 75.96 | 75.03 | [59.42–91.67] |
| | BioMedCLIP | A4C | 25 | 64.00 | 63.53 | [44.00–80.00] | 60.06 | 59.84 | [44.73–75.01] | 57.14 | 55.58 | [35.89–77.72] |
| | DINOv3 | A4C | 25 | 72.00 | 71.54 | [52.00–88.00] | 70.13 | 69.87 | [51.84–86.68] | 70.29 | 68.99 | [50.00–86.63] |
| | SigLIP2 | A4C | 25 | 72.00 | 71.54 | [52.00–88.00] | 70.13 | 69.87 | [51.84–86.68] | 70.29 | 68.99 | [50.00–86.63] |
| | EchoCLIP | combined | 50 | 74.00 | 74.05 | [62.00–86.00] | 74.20 | 74.24 | [61.20–86.04] | 73.99 | 73.64 | [60.00–85.86] |
| | EchoPrime | combined | 50 | 80.00 | 80.24 | [70.00–90.00] | 80.13 | 80.35 | [69.80–90.92] | 80.00 | 79.90 | [68.78–90.00] |
| | PanEcho | combined | 50 | 70.00 | 70.55 | [57.95–82.00] | 70.51 | 71.07 | [58.33–83.28] | 69.70 | 69.83 | [55.93–81.99] |
| | EchoFM | combined | 50 | 72.00 | 71.87 | [60.00–84.00] | 72.44 | 72.33 | [59.93–83.57] | 71.82 | 71.28 | [57.98–83.04] |
| | BioMedCLIP | combined | 50 | 60.00 | 60.04 | [46.00–74.00] | 58.81 | 58.82 | [48.00–70.00] | 55.44 | 54.90 | [40.26–68.82] |
| | DINOv3 | combined | 50 | 76.00 | 75.90 | [64.00–86.00] | 75.32 | 75.21 | [64.00–85.71] | 75.00 | 74.48 | [62.49–85.95] |
| | SigLIP2 | combined | 50 | 76.00 | 75.90 | [64.00–86.00] | 75.32 | 75.21 | [64.00–85.71] | 75.00 | 74.48 | [62.49–85.95] |

Table 11: Zero-shot and linear probing performance on TMED-2 dataset. The *View* column indicates the view of the ground truth: TMED-2 includes A2C, A4C, PLAX, PSAX, and Other. In the main paper, results are reported at the study level, but here we additionally provide view-specific results.

| Model | View | $n$ | Accuracy (%) | | | Balanced Accuracy (%) | | | F1 (%) | | |
|---|---|---|---|---|---|---|---|---|---|---|---|
| | | | Raw | Mean | CI | Raw | Mean | CI | Raw | Mean | CI |
| EchoCLIP | A2C | 297 | 94.95 | 94.93 | [92.26–97.31] | 50.00 | 50.00 | [50.00–50.00] | 48.70 | 48.70 | [47.99–49.32] |
| PanEcho | A2C | 297 | 84.18 | 84.16 | [80.13–88.22] | 66.42 | 65.95 | [52.78–79.18] | 57.07 | 56.72 | [49.77–64.29] |
| BioMedCLIP | A2C | 297 | 94.95 | 94.93 | [92.26–97.31] | 50.00 | 50.00 | [50.00–50.00] | 48.70 | 48.70 | [47.99–49.32] |
| DINOv3 | A2C | 297 | 94.95 | 94.93 | [92.26–97.31] | 50.00 | 50.00 | [50.00–50.00] | 48.70 | 48.70 | [47.99–49.32] |
| SigLIP2 | A2C | 297 | 5.05 | 5.07 | [2.69–7.74] | 50.00 | 50.00 | [50.00–50.00] | 4.81 | 4.81 | [2.62–7.19] |
| EchoCLIP | A4C | 430 | 94.19 | 94.25 | [92.09–96.28] | 50.00 | 50.00 | [50.00–50.00] | 48.50 | 48.52 | [47.94–49.05] |
| PanEcho | A4C | 430 | 82.79 | 82.84 | [78.84–86.28] | 75.85 | 75.75 | [65.91–85.75] | 60.82 | 60.61 | [54.33–66.72] |
| BioMedCLIP | A4C | 430 | 94.19 | 94.25 | [92.09–96.28] | 50.00 | 50.00 | [50.00–50.00] | 48.50 | 48.52 | [47.94–49.05] |
| DINOv3 | A4C | 430 | 94.19 | 94.25 | [92.09–96.28] | 50.00 | 50.00 | [50.00–50.00] | 48.50 | 48.52 | [47.94–49.05] |
| SigLIP2 | A4C | 430 | 5.81 | 5.75 | [3.72–7.91] | 50.00 | 50.00 | [50.00–50.00] | 5.49 | 5.43 | [3.59–7.33] |
| EchoCLIP | PLAX | 994 | 74.75 | 74.74 | [71.93–77.46] | 50.00 | 50.00 | [50.00–50.00] | 42.77 | 42.77 | [41.84–43.65] |
| PanEcho | PLAX | 994 | 70.82 | 70.83 | [68.01–73.74] | 55.03 | 55.03 | [52.10–57.94] | 55.12 | 55.10 | [51.65–58.45] |
| BioMedCLIP | PLAX | 994 | 74.75 | 74.74 | [71.93–77.46] | 50.00 | 50.00 | [50.00–50.00] | 42.77 | 42.77 | [41.84–43.65] |
| DINOv3 | PLAX | 994 | 74.75 | 74.74 | [71.93–77.46] | 50.00 | 50.00 | [50.00–50.00] | 42.77 | 42.77 | [41.84–43.65] |
| SigLIP2 | PLAX | 994 | 25.25 | 25.26 | [22.54–28.07] | 50.00 | 50.00 | [50.00–50.00] | 20.16 | 20.16 | [18.39–21.92] |
| EchoCLIP | PSAX | 383 | 83.29 | 83.32 | [79.63–86.68] | 50.00 | 50.00 | [50.00–50.00] | 45.44 | 45.44 | [44.33–46.43] |
| PanEcho | PSAX | 383 | 78.59 | 78.61 | [74.41–82.51] | 60.29 | 60.27 | [54.36–66.34] | 60.55 | 60.43 | [54.38–66.41] |
| BioMedCLIP | PSAX | 383 | 83.29 | 83.32 | [79.63–86.68] | 50.00 | 50.00 | [50.00–50.00] | 45.44 | 45.44 | [44.33–46.43] |
| DINOv3 | PSAX | 383 | 83.29 | 83.32 | [79.63–86.68] | 50.00 | 50.00 | [50.00–50.00] | 45.44 | 45.44 | [44.33–46.43] |
| SigLIP2 | PSAX | 383 | 16.71 | 16.68 | [13.32–20.37] | 50.00 | 50.00 | [50.00–50.00] | 14.32 | 14.28 | [11.75–16.92] |
| EchoCLIP | Other | 1498 | 100.00 | 100.00 | [100.00–100.00] | 100.00 | 100.00 | [100.00–100.00] | 100.00 | 100.00 | [100.00–100.00] |
| PanEcho | Other | 1498 | 85.85 | 85.82 | [84.04–87.45] | 85.85 | 85.82 | [84.04–87.45] | 92.39 | 92.37 | [91.33–93.31] |
| BioMedCLIP | Other | 1498 | 100.00 | 100.00 | [100.00–100.00] | 100.00 | 100.00 | [100.00–100.00] | 100.00 | 100.00 | [100.00–100.00] |
| DINOv3 | Other | 1498 | 100.00 | 100.00 | [100.00–100.00] | 100.00 | 100.00 | [100.00–100.00] | 100.00 | 100.00 | [100.00–100.00] |
| SigLIP2 | Other | 1498 | 0.00 | 0.00 | [0.00–0.00] | 0.00 | 0.00 | [0.00–0.00] | 0.00 | 0.00 | [0.00–0.00] |
| EchoCLIP | study-level | 119 | 78.99 | 78.85 | [71.43–86.55] | 50.00 | 50.00 | [50.00–50.00] | 44.13 | 44.06 | [41.67–46.40] |
| EchoPrime | study-level | 119 | 78.99 | 78.85 | [71.43–86.55] | 50.00 | 50.00 | [50.00–50.00] | 44.13 | 44.06 | [41.67–46.40] |
| PanEcho | study-level | 119 | 73.95 | 73.95 | [66.37–81.51] | 58.55 | 58.63 | [49.32–68.32] | 58.91 | 58.73 | [48.91–68.80] |
| BioMedCLIP | study-level | 119 | 78.99 | 78.85 | [71.43–86.55] | 50.00 | 50.00 | [50.00–50.00] | 44.13 | 44.06 | [41.67–46.40] |
| DINOv3 | study-level | 119 | 78.99 | 78.85 | [71.43–86.55] | 50.00 | 50.00 | [50.00–50.00] | 44.13 | 44.06 | [41.67–46.40] |
| SigLIP2 | study-level | 119 | 21.01 | 21.15 | [13.45–28.57] | 50.00 | 50.00 | [50.00–50.00] | 17.36 | 17.38 | [11.85–22.22] |

Table 12: Zero-shot and linear probing performance on SegRWMA dataset. The *View* column indicates the view of the ground truth: SegRWMA includes A2C, A3C, and A4C with 95% confidence intervals.

| Setting | Model | View | $n$ | Accuracy (%) | | | Balanced Accuracy (%) | | | F1 (%) | | |
|---|---|---|---|---|---|---|---|---|---|---|---|---|
| | | | | Raw | Mean | CI | Raw | Mean | CI | Raw | Mean | CI |
| Zero-shot | EchoCLIP | A2C | 48 | 43.75 | 43.74 | [29.17–58.33] | 46.59 | 47.03 | [18.48–74.44] | 35.68 | 35.95 | [24.43–50.03] |
| | PanEcho | A2C | 48 | 33.33 | 33.56 | [20.83–47.92] | 52.27 | 51.77 | [15.22–69.57] | 30.31 | 30.50 | [19.58–43.79] |
| | BioMedCLIP | A2C | 48 | 43.75 | 43.84 | [29.17–58.33] | 57.95 | 58.19 | [24.99–76.14] | 37.66 | 37.92 | [25.81–54.55] |
| | DINOv3 | A2C | 48 | 91.67 | 91.70 | [83.33–97.92] | 50.00 | 50.95 | [50.00–50.00] | 47.83 | 48.76 | [45.45–49.47] |
| | SigLIP2 | A2C | 48 | 89.58 | 89.81 | [81.25–97.92] | 48.86 | 49.89 | [46.51–50.00] | 47.25 | 48.23 | [44.83–49.47] |
| | EchoCLIP | A4C | 56 | 14.29 | 14.39 | [7.14–25.00] | 53.85 | 53.12 | [50.93–57.69] | 14.29 | 14.31 | [7.02–24.90] |
| | PanEcho | A4C | 56 | 21.43 | 21.28 | [10.71–32.14] | 46.15 | 45.27 | [10.36–62.73] | 20.52 | 20.36 | [10.60–31.36] |
| | BioMedCLIP | A4C | 56 | 7.14 | 7.34 | [1.79–14.29] | 50.00 | 49.25 | [50.00–50.00] | 6.67 | 6.75 | [1.75–12.50] |
| | DINOv3 | A4C | 56 | 92.86 | 92.66 | [85.71–98.21] | 50.00 | 50.75 | [50.00–50.00] | 48.15 | 48.83 | [46.15–49.55] |
| | SigLIP2 | A4C | 56 | 89.29 | 89.10 | [80.36–96.43] | 48.08 | 48.81 | [45.19–50.00] | 47.17 | 47.83 | [44.55–49.09] |
| | EchoCLIP | A3C | 52 | 42.31 | 42.14 | [28.85–55.77] | 57.29 | 56.54 | [21.00–74.52] | 36.27 | 36.14 | [25.36–49.03] |
| | PanEcho | A3C | 52 | 25.00 | 25.03 | [13.46–36.54] | 59.38 | 58.81 | [53.12–65.31] | 24.30 | 24.22 | [13.43–35.95] |
| | BioMedCLIP | A3C | 52 | 34.62 | 34.73 | [23.08–48.08] | 64.58 | 64.12 | [58.00–70.93] | 32.10 | 32.12 | [21.21–44.85] |
| | DINOv3 | A3C | 52 | 92.31 | 92.35 | [84.62–98.08] | 50.00 | 50.75 | [50.00–50.00] | 48.00 | 48.74 | [45.83–49.51] |
| | SigLIP2 | A3C | 52 | 92.31 | 92.30 | [84.62–98.08] | 72.92 | 73.18 | [46.94–99.00] | 72.92 | 71.56 | [46.94–97.03] |
| Linear Probing | EchoCLIP | A2C | 48 | 91.67 | 91.70 | [83.33–97.92] | 50.00 | 50.95 | [50.00–50.00] | 47.83 | 48.76 | [45.45–49.47] |
| | EchoPrime | A2C | 48 | 8.33 | 8.45 | [2.08–16.67] | 27.27 | 26.74 | [1.09–53.33] | 8.33 | 8.37 | [2.04–16.52] |
| | PanEcho | A2C | 48 | 91.67 | 91.55 | [83.33–97.92] | 72.73 | 73.26 | [46.67–98.91] | 72.73 | 71.37 | [46.67–96.80] |
| | BioMedCLIP | A2C | 48 | 91.67 | 91.70 | [83.33–97.92] | 50.00 | 50.95 | [50.00–50.00] | 47.83 | 48.76 | [45.45–49.47] |
| | DINOv3 | A2C | 48 | 91.67 | 91.70 | [83.33–97.92] | 50.00 | 50.95 | [50.00–50.00] | 47.83 | 48.76 | [45.45–49.47] |
| | SigLIP2 | A2C | 48 | 91.67 | 91.70 | [83.33–97.92] | 50.00 | 50.95 | [50.00–50.00] | 47.83 | 48.76 | [45.45–49.47] |
| | EchoFM | A2C | 48 | 91.67 | 91.70 | [83.33–97.92] | 50.00 | 50.95 | [50.00–50.00] | 47.83 | 48.76 | [45.45–49.47] |
| | EchoCLIP | A4C | 56 | 92.86 | 92.66 | [85.71–98.21] | 50.00 | 50.75 | [50.00–50.00] | 48.15 | 48.83 | [46.15–49.55] |
| | EchoPrime | A4C | 56 | 92.86 | 92.66 | [85.71–98.21] | 50.00 | 50.75 | [50.00–50.00] | 48.15 | 48.83 | [46.15–49.55] |
| | PanEcho | A4C | 56 | 92.86 | 92.88 | [85.71–98.21] | 61.54 | 62.32 | [47.27–98.24] | 64.78 | 63.94 | [46.67–92.38] |
| | BioMedCLIP | A4C | 56 | 92.86 | 92.66 | [85.71–98.21] | 50.00 | 50.75 | [50.00–50.00] | 48.15 | 48.83 | [46.15–49.55] |
| | DINOv3 | A4C | 56 | 92.86 | 92.66 | [85.71–98.21] | 50.00 | 50.75 | [50.00–50.00] | 48.15 | 48.83 | [46.15–49.55] |
| | SigLIP2 | A4C | 56 | 92.86 | 92.66 | [85.71–98.21] | 50.00 | 50.75 | [50.00–50.00] | 48.15 | 48.83 | [46.15–49.55] |
| | EchoFM | A4C | 56 | 92.86 | 92.66 | [85.71–98.21] | 50.00 | 50.75 | [50.00–50.00] | 48.15 | 48.83 | [46.15–49.55] |
| | EchoCLIP | A3C | 52 | 92.31 | 92.35 | [84.62–98.08] | 50.00 | 50.75 | [50.00–50.00] | 48.00 | 48.74 | [45.83–49.51] |
| | EchoPrime | A3C | 52 | 94.23 | 94.21 | [86.54–100.00] | 62.50 | 62.56 | [50.00–100.00] | 68.48 | 65.51 | [47.47–100.00] |
| | PanEcho | A3C | 52 | 90.38 | 90.44 | [80.77–98.08] | 48.96 | 49.70 | [46.74–50.00] | 47.47 | 48.21 | [44.68–49.51] |
| | EchoFM | A3C | 52 | 92.31 | 92.35 | [84.62–98.08] | 50.00 | 50.75 | [50.00–50.00] | 48.00 | 48.74 | [45.83–49.51] |
| | BioMedCLIP | A3C | 52 | 92.31 | 92.35 | [84.62–98.08] | 50.00 | 50.75 | [50.00–50.00] | 48.00 | 48.74 | [45.83–49.51] |
| | DINOv3 | A3C | 52 | 92.31 | 92.35 | [84.62–98.08] | 50.00 | 50.75 | [50.00–50.00] | 48.00 | 48.74 | [45.83–49.51] |
| | SigLIP2 | A3C | 52 | 92.31 | 92.35 | [84.62–98.08] | 50.00 | 50.75 | [50.00–50.00] | 48.00 | 48.74 | [45.83–49.51] |

Table 13: View classification performance with 95% confidence intervals.

| Dataset | Model | $n$ | Accuracy (%) Raw | Mean | CI | Balanced Accuracy (%) Raw | Mean | CI | F1 (%) Raw | Mean | CI |
|---|---|---|---|---|---|---|---|---|---|---|---|
| CAMUS | EchoCLIP | 100 | 49.00 | 48.76 | [39.00–58.00] | 49.00 | 48.97 | [46.88–50.00] | 33.11 | 32.92 | [28.15–37.01] |
| | EchoPrime | 100 | 10.00 | 10.04 | [5.00–16.00] | 10.00 | 10.02 | [4.89–16.00] | 16.39 | 16.22 | [8.33–24.51] |
| | BioMedCLIP | 100 | 13.00 | 13.06 | [7.00–20.00] | 13.00 | 13.04 | [7.00–19.52] | 17.02 | 16.85 | [8.98–25.95] |
| | DINOv3 | 100 | 0.00 | 0.00 | [0.00–0.00] | 0.00 | 0.00 | [0.00–0.00] | 0.00 | 0.00 | [0.00–0.00] |
| | SigLIP2 | 100 | 4.00 | 4.04 | [1.00–8.00] | 4.00 | 4.02 | [0.93–8.49] | 6.67 | 6.61 | [1.67–12.91] |
| EchoNet Dynamic | EchoCLIP | 1277 | 4.46 | 4.45 | [3.37–5.72] | 4.46 | 4.45 | [3.37–5.72] | 8.55 | 8.52 | [6.52–10.81] |
| | EchoPrime | 1277 | 97.02 | 97.03 | [96.08–97.96] | 97.02 | 97.03 | [96.08–97.96] | 98.49 | 98.49 | [98.00–98.97] |
| | BioMedCLIP | 1277 | 0.00 | 0.00 | [0.00–0.00] | 0.00 | 0.00 | [0.00–0.00] | 0.00 | 0.00 | [0.00–0.00] |
| | DINOv3 | 1277 | 0.16 | 0.16 | [0.00–0.39] | 0.16 | 0.16 | [0.00–0.39] | 0.31 | 0.31 | [0.00–0.78] |
| | SigLIP2 | 1277 | 77.45 | 77.41 | [75.18–79.87] | 77.45 | 77.41 | [75.18–79.87] | 87.29 | 87.26 | [85.83–88.81] |
| EchoNet Pediatric | EchoCLIP | 658 | 17.17 | 17.15 | [14.13–20.06] | 16.48 | 16.45 | [13.67–19.38] | 20.95 | 20.89 | [17.59–24.18] |
| | EchoPrime | 658 | 67.78 | 67.75 | [64.13–71.28] | 66.58 | 66.55 | [62.88–70.39] | 79.53 | 79.48 | [76.64–82.32] |
| | BioMedCLIP | 658 | 0.00 | 0.00 | [0.00–0.00] | 0.00 | 0.00 | [0.00–0.00] | 0.00 | 0.00 | [0.00–0.00] |
| | DINOv3 | 658 | 46.81 | 46.73 | [42.86–50.30] | 39.85 | 39.79 | [37.53–41.83] | 35.82 | 35.75 | [33.83–37.60] |
| | SigLIP2 | 658 | 40.43 | 40.38 | [36.63–44.38] | 44.04 | 44.00 | [40.63–47.70] | 45.32 | 45.20 | [41.43–49.50] |
| EchoNet LVH | EchoCLIP | 340 | 0.88 | 0.89 | [0.00–2.06] | 0.88 | 0.89 | [0.00–2.06] | 1.75 | 1.76 | [0.00–4.03] |
| | EchoPrime | 340 | 97.35 | 97.37 | [95.59–98.82] | 97.35 | 97.37 | [95.59–98.82] | 98.66 | 98.66 | [97.74–99.41] |
| | BioMedCLIP | 340 | 0.29 | 0.28 | [0.00–0.88] | 0.29 | 0.28 | [0.00–0.88] | 0.59 | 0.57 | [0.00–1.75] |
| | DINOv3 | 340 | 0.00 | 0.00 | [0.00–0.00] | 0.00 | 0.00 | [0.00–0.00] | 0.00 | 0.00 | [0.00–0.00] |
| | SigLIP2 | 340 | 17.06 | 17.03 | [13.24–20.88] | 17.06 | 17.03 | [13.24–20.88] | 29.15 | 29.05 | [23.38–34.55] |
| CardiacNet | $biomedclip_a sd$ | 47 | 12.77 | 13.01 | [4.26–23.40] | 12.77 | 13.01 | [4.26–23.40] | 22.64 | 22.70 | [8.16–37.93] |
| | $biomedclip_p ah$ | 106 | 83.02 | 83.01 | [75.47–89.62] | 83.02 | 83.01 | [75.47–89.62] | 90.72 | 90.67 | [86.02–94.53] |
| | $dinov3_a sd$ | 47 | 0.00 | 0.00 | [0.00–0.00] | 0.00 | 0.00 | [0.00–0.00] | 0.00 | 0.00 | [0.00–0.00] |
| | $dinov3_p ah$ | 106 | 0.00 | 0.00 | [0.00–0.00] | 0.00 | 0.00 | [0.00–0.00] | 0.00 | 0.00 | [0.00–0.00] |
| | $echoclip_a sd$ | 47 | 17.02 | 16.96 | [6.38–27.66] | 17.02 | 16.96 | [6.38–27.66] | 29.09 | 28.64 | [12.00–43.33] |
| | $echoclip_p ah$ | 106 | 15.09 | 15.12 | [7.55–22.64] | 15.09 | 15.12 | [7.55–22.64] | 26.23 | 26.10 | [14.04–36.92] |
| | $echoprime_a sd$ | 47 | 59.57 | 59.31 | [44.63–72.34] | 59.57 | 59.31 | [44.63–72.34] | 74.67 | 74.19 | [61.71–83.95] |
| | $echoprime_p ah$ | 106 | 70.75 | 70.66 | [61.32–79.25] | 70.75 | 70.66 | [61.32–79.25] | 82.87 | 82.73 | [76.02–88.42] |
| | $sigclip_a sd$ | 47 | 36.17 | 36.29 | [23.40–51.06] | 36.17 | 36.29 | [23.40–51.06] | 53.12 | 52.85 | [37.93–67.61] |
| | $sigclip_p ah$ | 106 | 38.68 | 39.06 | [30.17–48.13] | 38.68 | 39.06 | [30.17–48.13] | 55.78 | 56.00 | [46.35–64.99] |
| HMC-QU | EchoCLIP | 50 | 46.00 | 45.90 | [32.00–60.00] | 46.00 | 45.95 | [39.99–50.00] | 34.33 | 34.10 | [27.12–40.01] |
| | EchoPrime | 50 | 82.00 | 81.92 | [70.00–92.00] | 82.00 | 81.90 | [70.14–92.00] | 88.19 | 87.92 | [79.02–94.89] |
| | BioMedCLIP | 50 | 48.00 | 48.05 | [34.00–62.00] | 48.00 | 48.08 | [34.17–62.66] | 47.67 | 47.16 | [33.33–61.98] |
| | DINOv3 | 50 | 0.00 | 0.00 | [0.00–0.00] | 0.00 | 0.00 | [0.00–0.00] | 0.00 | 0.00 | [0.00–0.00] |
| | SigLIP2 | 50 | 36.00 | 36.08 | [24.00–50.00] | 36.00 | 36.11 | [22.90–49.36] | 41.37 | 40.83 | [26.92–56.02] |
| TMED-2 | EchoCLIP | 3602 | 15.74 | 15.72 | [14.60–16.94] | 16.57 | 16.54 | [15.20–17.99] | 14.25 | 14.22 | [13.10–15.42] |
| | EchoPrime | 3602 | 61.13 | 61.12 | [59.49–62.69] | 76.09 | 76.06 | [74.84–77.20] | 62.86 | 62.84 | [61.33–64.32] |
| | BioMedCLIP | 3602 | 24.99 | 24.97 | [23.63–26.40] | 34.90 | 34.85 | [33.04–36.73] | 26.37 | 26.34 | [24.98–27.74] |
| | DINOv3 | 3602 | 6.11 | 6.12 | [5.39–6.94] | 11.69 | 11.69 | [10.63–12.82] | 4.89 | 4.90 | [4.11–5.77] |
| | SigLIP2 | 3602 | 20.63 | 20.61 | [19.32–21.99] | 28.22 | 28.20 | [26.73–29.66] | 16.17 | 16.15 | [15.03–17.30] |
| SegRWMA | EchoCLIP | 156 | 25.00 | 24.99 | [18.59–31.41] | 23.41 | 23.33 | [18.61–27.91] | 16.86 | 16.71 | [12.16–21.37] |
| | EchoPrime | 156 | 10.26 | 10.27 | [5.77–15.38] | 10.36 | 10.34 | [5.86–15.26] | 15.79 | 15.58 | [8.95–22.48] |
| | BioMedCLIP | 156 | 18.59 | 18.66 | [12.18–25.00] | 19.54 | 19.54 | [13.58–25.41] | 18.41 | 18.33 | [12.97–23.57] |
| | DINOv3 | 156 | 0.00 | 0.00 | [0.00–0.00] | 0.00 | 0.00 | [0.00–0.00] | 0.00 | 0.00 | [0.00–0.00] |
| | SigLIP2 | 156 | 1.28 | 1.25 | [0.00–3.21] | 1.29 | 1.25 | [0.00–3.21] | 2.43 | 2.32 | [0.00–5.83] |

# H   SUBGROUP ROBUSTNESS

We assess subgroup robustness by evaluating model performance across demographic and acquisition-related groups. For each group, we report standard regression metrics (MAE, nMAE, MSE, RMSE) along with the sample size. To capture disparities, we add a $\Delta$(max–min) row per grouping, summarizing the gap between the best- and worst-performing groups. This follows the definition of $\Delta$ metrics used in classification bias analysis (e.g., $\Delta$AUC Jin et al. (2024)). Together, per-group results show where models underperform, while $\Delta$ highlights overall spread. Complete subgroup tables are provided below.

Table 14: Subgroup results by age with per-group raw metrics and a $\Delta$ (max$-$min) summary on CAMUS

| Model | View | Group | $n$ | MAE | NMAE (%) | MSE | RMSE | $R^2$ (%) | Pearson $r$ (%) | Spearman $\rho$ (%) |
|---|---|---|---|---|---|---|---|---|---|---|
| EchoCLIP | A2C | 45 | 5 | 13.98 | 21.84 | 339.19 | 18.42 | 23.78 | 54.18 | 30.00 |
| | | 46–65 | 20 | 9.41 | 26.13 | 115.02 | 10.72 | -7.87 | 13.21 | 10.85 |
| | | 66–80 | 20 | 8.66 | 25.47 | 108.84 | 10.43 | -66.71 | 40.09 | 41.01 |
| | | $\Delta$ (max$-$min) | 45 | 5.32 | 4.29 | 230.35 | 7.99 | 90.49 | 40.97 | 30.16 |
| EchoCLIP | A4C | 45 | 5 | 15.80 | 24.69 | 422.45 | 20.55 | 5.08 | 47.79 | 70.00 |
| | | 46–65 | 20 | 10.76 | 29.89 | 184.87 | 13.60 | -73.38 | -22.60 | -17.41 |
| | | 66–80 | 20 | 8.39 | 24.66 | 102.53 | 10.13 | -57.04 | 48.52 | 39.25 |
| | | $\Delta$ (max$-$min) | 45 | 7.41 | 5.23 | 319.92 | 10.42 | 78.46 | 71.12 | 87.41 |
| EchoCLIP | combined | 45 | 5 | 14.89 | 23.27 | 369.61 | 19.23 | 16.95 | 53.48 | 40.00 |
| | | 46–65 | 20 | 9.87 | 27.42 | 136.48 | 11.68 | -28.00 | -9.44 | -3.99 |
| | | 66–80 | 20 | 8.27 | 24.33 | 97.57 | 9.88 | -49.44 | 48.26 | 40.72 |
| | | $\Delta$ (max$-$min) | 45 | 6.62 | 4.15 | 272.04 | 9.35 | 66.39 | 62.92 | 44.71 |
| EchoPrime | combined | 45 | 5 | 24.00 | 37.51 | 853.93 | 29.22 | -91.88 | 79.39 | 70.00 |
| | | 46–65 | 20 | 13.82 | 38.38 | 288.40 | 16.98 | -170.47 | 38.72 | 39.86 |
| | | 66–80 | 19 | 11.70 | 34.40 | 188.15 | 13.72 | -185.36 | 28.34 | 39.33 |
| | | $\Delta$ (max$-$min) | 44 | 12.30 | 3.98 | 665.78 | 15.50 | 93.48 | 51.05 | 30.67 |
| PanEcho | A2C | 45 | 5 | 16.79 | 26.24 | 414.35 | 20.36 | 6.90 | 71.63 | 60.00 |
| | | 46–65 | 20 | 11.08 | 30.78 | 187.46 | 13.69 | -75.81 | 60.20 | 64.96 |
| | | 66–80 | 20 | 10.78 | 31.71 | 177.42 | 13.32 | -171.74 | 26.50 | 24.27 |
| | | $\Delta$ (max$-$min) | 45 | 6.01 | 5.47 | 236.93 | 7.04 | 178.64 | 45.13 | 40.69 |
| PanEcho | A4C | 45 | 5 | 15.59 | 24.36 | 315.55 | 17.76 | 29.10 | 71.92 | 60.00 |
| | | 46–65 | 20 | 12.87 | 35.74 | 268.42 | 16.38 | -151.74 | 43.93 | 48.76 |
| | | 66–80 | 20 | 10.27 | 30.22 | 171.09 | 13.08 | -162.04 | 27.29 | 35.58 |
| | | $\Delta$ (max$-$min) | 45 | 5.32 | 11.38 | 144.46 | 4.68 | 191.14 | 44.63 | 24.42 |
| PanEcho | combined | 45 | 5 | 16.19 | 25.30 | 352.53 | 18.78 | 20.79 | 73.16 | 60.00 |
| | | 46–65 | 20 | 11.77 | 32.70 | 219.18 | 14.80 | -105.55 | 54.90 | 61.64 |
| | | 66–80 | 20 | 10.06 | 29.59 | 167.84 | 12.96 | -157.07 | 28.21 | 25.25 |
| | | $\Delta$ (max$-$min) | 45 | 6.13 | 7.40 | 184.69 | 5.82 | 177.86 | 44.95 | 36.39 |
| BioMedCLIP | A2C | 45 | 5 | 30.20 | 47.19 | 1251.23 | 35.37 | -181.15 | -77.99 | -70.00 |
| | | 46–65 | 20 | 20.30 | 56.39 | 495.97 | 22.27 | -365.14 | 47.31 | 53.17 |
| | | 66–80 | 20 | 12.08 | 35.53 | 229.56 | 15.15 | -251.60 | -29.83 | -33.86 |
| | | $\Delta$ (max$-$min) | 45 | 18.12 | 20.86 | 1021.67 | 20.22 | 183.99 | 125.30 | 123.17 |
| BioMedCLIP | A4C | 45 | 5 | 36.32 | 56.75 | 1692.82 | 41.14 | -280.37 | -86.94 | -70.00 |
| | | 46–65 | 20 | 21.39 | 59.42 | 566.88 | 23.81 | -431.65 | 16.14 | 18.61 |
| | | 66–80 | 20 | 13.88 | 40.84 | 270.14 | 16.44 | -313.76 | 8.56 | 12.66 |
| | | $\Delta$ (max$-$min) | 45 | 22.44 | 18.58 | 1422.68 | 24.70 | 151.28 | 103.08 | 88.61 |
| BioMedCLIP | combined | 45 | 5 | 33.26 | 51.97 | 1455.58 | 38.15 | -227.07 | -87.68 | -60.00 |
| | | 46–65 | 20 | 20.84 | 57.90 | 524.34 | 22.90 | -391.75 | 41.95 | 37.91 |
| | | 66–80 | 20 | 12.96 | 38.11 | 245.06 | 15.65 | -275.33 | -7.23 | -9.50 |
| | | $\Delta$ (max$-$min) | 45 | 20.30 | 19.79 | 1210.52 | 22.50 | 164.68 | 129.63 | 97.91 |
| DINOv3 | A2C | 45 | 5 | 22.99 | 35.92 | 681.69 | 26.11 | -53.18 | 62.38 | 80.00 |
| | | 46–65 | 20 | 9.64 | 26.79 | 172.84 | 13.15 | -62.10 | -33.73 | -14.32 |
| | | 66–80 | 20 | 6.50 | 19.12 | 65.30 | 8.08 | -0.01 | -0.54 | -14.78 |
| | | $\Delta$ (max$-$min) | 45 | 16.49 | 16.80 | 616.39 | 18.03 | 62.09 | 96.11 | 94.78 |
| DINOv3 | A4C | 45 | 5 | 22.99 | 35.92 | 681.77 | 26.11 | -53.19 | -41.07 | -40.00 |
| | | 46–65 | 20 | 9.65 | 26.79 | 172.86 | 13.15 | -62.12 | -53.10 | -44.31 |
| | | 66–80 | 20 | 6.50 | 19.12 | 65.29 | 8.08 | -0.00 | 16.60 | 14.85 |
| | | $\Delta$ (max$-$min) | 45 | 16.49 | 16.80 | 616.48 | 18.03 | 62.12 | 69.70 | 59.16 |
| DINOv3 | combined | 45 | 5 | 22.99 | 35.92 | 681.73 | 26.11 | -53.18 | 11.22 | 30.00 |
| | | 46–65 | 20 | 9.64 | 26.79 | 172.85 | 13.15 | -62.11 | -48.95 | -38.36 |
| | | 66–80 | 20 | 6.50 | 19.12 | 65.29 | 8.08 | -0.01 | 8.19 | 5.35 |
| | | $\Delta$ (max$-$min) | 45 | 16.49 | 16.80 | 616.44 | 18.03 | 62.10 | 60.17 | 68.36 |
| SigLIP2 | A2C | 45 | 5 | 20.00 | 31.25 | 553.24 | 23.52 | -24.31 | -67.48 | -60.00 |
| | | 46–65 | 20 | 8.55 | 23.75 | 116.54 | 10.80 | -9.30 | -0.23 | -2.19 |
| | | 66–80 | 20 | 7.60 | 22.36 | 89.32 | 9.45 | -36.81 | -15.65 | -30.46 |
| | | $\Delta$ (max$-$min) | 45 | 12.40 | 8.89 | 463.92 | 14.07 | 27.51 | 67.25 | 57.81 |
| SigLIP2 | A4C | 45 | 5 | 20.00 | 31.25 | 553.30 | 23.52 | -24.33 | -68.40 | -60.00 |
| | | 46–65 | 20 | 8.55 | 23.75 | 116.55 | 10.80 | -9.30 | -18.48 | -23.13 |
| | | 66–80 | 20 | 7.60 | 22.36 | 89.32 | 9.45 | -36.80 | 23.32 | 17.57 |
| | | $\Delta$ (max$-$min) | 45 | 12.40 | 8.89 | 463.98 | 14.07 | 27.50 | 91.72 | 77.57 |
| SigLIP2 | combined | 45 | 5 | 20.00 | 31.25 | 553.27 | 23.52 | -24.32 | -68.18 | -60.00 |
| | | 46–65 | 20 | 8.55 | 23.75 | 116.55 | 10.80 | -9.30 | -12.11 | -17.94 |
| | | 66–80 | 20 | 7.60 | 22.36 | 89.32 | 9.45 | -36.80 | 2.59 | 11.23 |
| | | $\Delta$ (max$-$min) | 45 | 12.40 | 8.89 | 463.95 | 14.07 | 27.50 | 70.77 | 71.23 |

Table 15: Subgroup analysis by gender with per-group raw metrics and a $\Delta$ (max$-$min) summary on CAMUS

| Model | View | Group | $n$ | MAE | NMAE (%) | MSE | RMSE | $R^2$ (%) | Pearson $r$ (%) | Spearman $\rho$ (%) |
|---|---|---|---|---|---|---|---|---|---|---|
| EchoCLIP | A2C | F | 12 | 11.48 | 26.69 | 170.37 | 13.05 | -21.48 | 14.54 | 16.81 |
| | | M | 38 | 9.34 | 15.06 | 143.79 | 11.99 | 5.59 | 36.04 | 34.25 |
| | | $\Delta$ (max$-$min) | 50 | 2.14 | 11.63 | 26.58 | 1.06 | 27.07 | 21.50 | 17.44 |
| EchoCLIP | A4C | F | 12 | 8.90 | 20.70 | 115.87 | 10.76 | 17.38 | 49.19 | 62.46 |
| | | M | 38 | 10.89 | 17.57 | 197.99 | 14.07 | -30.00 | 3.53 | 0.81 |
| | | $\Delta$ (max$-$min) | 50 | 1.99 | 3.13 | 82.12 | 3.31 | 47.38 | 45.66 | 61.65 |
| EchoCLIP | combined | F | 12 | 9.70 | 22.57 | 125.21 | 11.19 | 10.72 | 42.49 | 30.82 |
| | | M | 38 | 9.87 | 15.92 | 159.71 | 12.64 | -4.86 | 21.50 | 13.39 |
| | | $\Delta$ (max$-$min) | 50 | 0.17 | 6.65 | 34.50 | 1.45 | 15.58 | 20.99 | 17.43 |
| EchoPrime | combined | F | 12 | 11.75 | 27.33 | 201.08 | 14.18 | -43.38 | 74.42 | 66.90 |
| | | M | 37 | 14.73 | 23.76 | 334.48 | 18.29 | -119.93 | 39.05 | 46.49 |
| | | $\Delta$ (max$-$min) | 49 | 2.98 | 3.57 | 133.40 | 4.11 | 76.55 | 35.37 | 20.41 |
| PanEcho | A2C | F | 12 | 10.08 | 23.44 | 164.03 | 12.81 | -16.96 | 60.71 | 59.54 |
| | | M | 38 | 12.41 | 20.01 | 228.27 | 15.11 | -49.88 | 61.20 | 54.04 |
| | | $\Delta$ (max$-$min) | 50 | 2.33 | 3.43 | 64.24 | 2.30 | 32.92 | 0.49 | 5.50 |
| PanEcho | A4C | F | 12 | 9.03 | 21.00 | 126.72 | 11.26 | 9.64 | 62.15 | 62.00 |
| | | M | 38 | 12.87 | 20.76 | 255.62 | 15.99 | -67.84 | 57.30 | 45.96 |
| | | $\Delta$ (max$-$min) | 50 | 3.84 | 0.24 | 128.90 | 4.73 | 77.48 | 4.85 | 16.04 |
| PanEcho | combined | F | 12 | 9.48 | 22.06 | 141.32 | 11.89 | -0.76 | 62.16 | 61.65 |
| | | M | 38 | 12.31 | 19.85 | 233.10 | 15.27 | -53.05 | 62.33 | 52.44 |
| | | $\Delta$ (max$-$min) | 50 | 2.83 | 2.21 | 91.78 | 3.38 | 52.29 | 0.17 | 9.21 |
| BioMedCLIP | A2C | F | 12 | 18.25 | 42.44 | 430.01 | 20.74 | -206.62 | -48.19 | -34.33 |
| | | M | 38 | 17.77 | 28.65 | 463.85 | 21.54 | -204.56 | 2.99 | 8.46 |
| | | $\Delta$ (max$-$min) | 50 | 0.48 | 13.79 | 33.84 | 0.80 | 2.06 | 51.18 | 42.79 |
| BioMedCLIP | A4C | F | 12 | 22.38 | 52.05 | 613.64 | 24.77 | -337.55 | -46.79 | -21.72 |
| | | M | 38 | 19.08 | 30.78 | 536.22 | 23.16 | -252.08 | -18.42 | -1.64 |
| | | $\Delta$ (max$-$min) | 50 | 3.30 | 21.27 | 77.42 | 1.61 | 85.47 | 28.37 | 20.08 |
| BioMedCLIP | combined | F | 12 | 20.32 | 47.25 | 515.04 | 22.69 | -267.25 | -49.75 | -32.57 |
| | | M | 38 | 18.41 | 29.69 | 493.32 | 22.21 | -223.91 | -10.22 | 10.97 |
| | | $\Delta$ (max$-$min) | 50 | 1.91 | 17.56 | 21.72 | 0.48 | 43.34 | 39.53 | 43.54 |
| DINOv3 | A2C | F | 12 | 10.58 | 24.61 | 161.13 | 12.69 | -14.89 | -9.88 | -24.17 |
| | | M | 38 | 9.65 | 15.57 | 182.40 | 13.51 | -19.76 | 5.44 | 0.82 |
| | | $\Delta$ (max$-$min) | 50 | 0.93 | 9.04 | 21.27 | 0.82 | 4.87 | 15.32 | 24.99 |
| DINOv3 | A4C | F | 12 | 10.58 | 24.61 | 161.13 | 12.69 | -14.89 | -9.40 | -6.66 |
| | | M | 38 | 9.65 | 15.57 | 182.41 | 13.51 | -19.77 | -11.66 | -7.43 |
| | | $\Delta$ (max$-$min) | 50 | 0.93 | 9.04 | 21.28 | 0.82 | 4.88 | 2.26 | 0.77 |
| DINOv3 | combined | F | 12 | 10.58 | 24.61 | 161.13 | 12.69 | -14.89 | -11.85 | 14.01 |
| | | M | 38 | 9.65 | 15.57 | 182.40 | 13.51 | -19.76 | -3.73 | -6.80 |
| | | $\Delta$ (max$-$min) | 50 | 0.93 | 9.04 | 21.27 | 0.82 | 4.87 | 8.12 | 20.81 |
| SigLIP2 | A2C | F | 12 | 9.75 | 22.67 | 140.43 | 11.85 | -0.13 | -32.29 | -41.33 |
| | | M | 38 | 9.13 | 14.73 | 152.56 | 12.35 | -0.17 | -14.16 | -16.06 |
| | | $\Delta$ (max$-$min) | 50 | 0.62 | 7.94 | 12.13 | 0.50 | 0.04 | 18.13 | 25.27 |
| SigLIP2 | A4C | F | 12 | 9.75 | 22.68 | 140.45 | 11.85 | -0.15 | -63.35 | -58.14 |
| | | M | 38 | 9.13 | 14.73 | 152.56 | 12.35 | -0.17 | -21.78 | -12.65 |
| | | $\Delta$ (max$-$min) | 50 | 0.62 | 7.95 | 12.11 | 0.50 | 0.02 | 41.57 | 45.49 |
| SigLIP2 | combined | F | 12 | 9.75 | 22.68 | 140.44 | 11.85 | -0.14 | -60.11 | -57.09 |
| | | M | 38 | 9.13 | 14.73 | 152.56 | 12.35 | -0.17 | -21.67 | -14.53 |
| | | $\Delta$ (max$-$min) | 50 | 0.62 | 7.95 | 12.12 | 0.50 | 0.03 | 38.44 | 42.56 |

Table 16: Subgroup results by image quality with per-group raw metrics and a $\Delta$ (max−min) summary on CAMUS

| Model | View | Group | $n$ | MAE | NMAE (%) | MSE | RMSE | $R^2$ (%) | Pearson $r$ (%) | Spearman $\rho$ (%) |
|---|---|---|---|---|---|---|---|---|---|---|
| EchoCLIP | A2C | Poor | 10 | 5.09 | 22.13 | 47.73 | 6.91 | 34.49 | 61.37 | 42.68 |
| | | Medium | 21 | 11.44 | 24.34 | 174.26 | 13.20 | -79.98 | -9.16 | 0.65 |
| | | Good | 19 | 10.60 | 16.56 | 177.47 | 13.32 | 18.33 | 42.97 | 40.23 |
| | | $\Delta$ (max−min) | 50 | 6.35 | 7.78 | 129.74 | 6.41 | 114.47 | 70.53 | 42.03 |
| EchoCLIP | A4C | Poor | 10 | 9.14 | 39.74 | 104.36 | 10.22 | -43.26 | 14.89 | 32.93 |
| | | Medium | 21 | 9.75 | 20.74 | 171.75 | 13.11 | -77.39 | -51.27 | -32.20 |
| | | Good | 19 | 11.82 | 18.47 | 224.40 | 14.98 | -3.27 | 24.77 | 10.80 |
| | | $\Delta$ (max−min) | 50 | 2.68 | 21.27 | 120.04 | 4.76 | 74.12 | 76.04 | 65.13 |
| EchoCLIP | combined | Poor | 10 | 6.28 | 27.33 | 55.06 | 7.42 | 24.42 | 50.39 | 52.44 |
| | | Medium | 21 | 10.48 | 22.29 | 161.80 | 12.72 | -67.12 | -37.96 | -22.53 |
| | | Good | 19 | 10.98 | 17.16 | 190.68 | 13.81 | 12.25 | 36.23 | 20.47 |
| | | $\Delta$ (max−min) | 50 | 4.70 | 10.17 | 135.62 | 6.39 | 91.54 | 88.35 | 74.97 |
| EchoPrime | combined | Poor | 10 | 18.93 | 82.32 | 430.81 | 20.76 | -491.36 | 25.48 | 29.27 |
| | | Medium | 21 | 12.53 | 26.67 | 234.83 | 15.32 | -142.55 | 38.87 | 57.07 |
| | | Good | 18 | 12.97 | 20.27 | 308.28 | 17.56 | -39.92 | 53.96 | 63.57 |
| | | $\Delta$ (max−min) | 49 | 6.40 | 62.05 | 195.98 | 5.44 | 451.44 | 28.48 | 34.30 |
| PanEcho | A2C | Poor | 10 | 15.22 | 66.16 | 336.49 | 18.34 | -361.89 | 29.20 | 6.10 |
| | | Medium | 21 | 10.40 | 22.13 | 160.52 | 12.67 | -65.79 | 61.09 | 50.67 |
| | | Good | 19 | 11.67 | 18.24 | 205.61 | 14.34 | 5.38 | 70.10 | 64.82 |
| | | $\Delta$ (max−min) | 50 | 4.82 | 47.92 | 175.97 | 5.67 | 367.27 | 40.90 | 58.72 |
| PanEcho | A4C | Poor | 10 | 13.96 | 60.71 | 290.71 | 17.05 | -299.05 | 39.57 | 13.41 |
| | | Medium | 21 | 10.84 | 23.07 | 196.30 | 14.01 | -102.75 | 48.07 | 40.42 |
| | | Good | 19 | 12.12 | 18.93 | 221.31 | 14.88 | -1.85 | 64.09 | 51.91 |
| | | $\Delta$ (max−min) | 50 | 3.12 | 41.78 | 94.41 | 3.04 | 297.20 | 24.52 | 38.50 |
| PanEcho | combined | Poor | 10 | 14.59 | 63.44 | 306.62 | 17.51 | -320.89 | 35.44 | 16.46 |
| | | Medium | 21 | 10.38 | 22.08 | 169.52 | 13.02 | -75.09 | 58.01 | 47.93 |
| | | Good | 19 | 11.46 | 17.91 | 206.71 | 14.38 | 4.87 | 69.55 | 64.21 |
| | | $\Delta$ (max−min) | 50 | 4.21 | 45.53 | 137.10 | 4.49 | 325.76 | 34.11 | 47.75 |
| BioMedCLIP | A2C | Poor | 10 | 22.18 | 96.43 | 535.94 | 23.15 | -635.68 | 67.17 | 64.03 |
| | | Medium | 21 | 14.39 | 30.61 | 284.88 | 16.88 | -194.23 | 10.68 | 8.36 |
| | | Good | 19 | 19.48 | 30.44 | 602.35 | 24.54 | -177.20 | -42.40 | -40.09 |
| | | $\Delta$ (max−min) | 50 | 7.79 | 65.99 | 317.47 | 7.66 | 458.48 | 109.57 | 104.12 |
| BioMedCLIP | A4C | Poor | 10 | 23.58 | 102.52 | 623.62 | 24.97 | -756.03 | 28.84 | 31.19 |
| | | Medium | 21 | 15.32 | 32.59 | 326.90 | 18.08 | -237.64 | -9.95 | -22.20 |
| | | Good | 19 | 22.96 | 35.87 | 770.47 | 27.76 | -254.57 | -50.34 | -29.60 |
| | | $\Delta$ (max−min) | 50 | 8.26 | 69.93 | 443.57 | 9.68 | 518.39 | 79.18 | 60.79 |
| BioMedCLIP | combined | Poor | 10 | 22.88 | 99.48 | 572.97 | 23.94 | -686.50 | 69.32 | 62.08 |
| | | Medium | 21 | 14.85 | 31.60 | 300.32 | 17.33 | -210.18 | -0.23 | -7.38 |
| | | Good | 19 | 21.19 | 33.12 | 678.44 | 26.05 | -212.22 | -53.08 | -33.55 |
| | | $\Delta$ (max−min) | 50 | 8.03 | 67.88 | 378.12 | 8.72 | 476.32 | 122.40 | 95.63 |
| DINOv3 | A2C | Poor | 10 | 11.49 | 49.94 | 204.80 | 14.31 | -181.12 | -19.58 | -0.61 |
| | | Medium | 21 | 6.67 | 14.19 | 101.56 | 10.08 | -4.90 | -12.23 | -11.36 |
| | | Good | 19 | 12.57 | 19.65 | 246.52 | 15.70 | -13.45 | 19.76 | 1.76 |
| | | $\Delta$ (max−min) | 50 | 5.90 | 35.75 | 144.96 | 5.62 | 176.22 | 39.34 | 13.12 |
| DINOv3 | A4C | Poor | 10 | 11.49 | 49.94 | 204.80 | 14.31 | -181.13 | -25.23 | 0.61 |
| | | Medium | 21 | 6.67 | 14.19 | 101.55 | 10.08 | -4.89 | 10.23 | 17.83 |
| | | Good | 19 | 12.57 | 19.65 | 246.56 | 15.70 | -13.46 | -19.03 | -40.05 |
| | | $\Delta$ (max−min) | 50 | 5.90 | 35.75 | 145.01 | 5.62 | 176.24 | 35.46 | 57.88 |
| DINOv3 | combined | Poor | 10 | 11.49 | 49.94 | 204.80 | 14.31 | -181.13 | -24.74 | 12.81 |
| | | Medium | 21 | 6.67 | 14.19 | 101.56 | 10.08 | -4.89 | 1.12 | 2.81 |
| | | Good | 19 | 12.57 | 19.65 | 246.54 | 15.70 | -13.46 | 4.84 | -7.73 |
| | | $\Delta$ (max−min) | 50 | 5.90 | 35.75 | 144.98 | 5.62 | 176.24 | 29.58 | 20.54 |
| SigLIP2 | A2C | Poor | 10 | 7.70 | 33.48 | 115.10 | 10.73 | -58.00 | -60.97 | -56.10 |
| | | Medium | 21 | 8.33 | 17.73 | 104.72 | 10.23 | -8.16 | 3.08 | -4.70 |
| | | Good | 19 | 11.16 | 17.43 | 217.48 | 14.75 | -0.09 | -21.48 | -38.30 |
| | | $\Delta$ (max−min) | 50 | 3.46 | 16.05 | 112.76 | 4.52 | 57.91 | 64.05 | 51.40 |
| SigLIP2 | A4C | Poor | 10 | 7.70 | 33.49 | 115.13 | 10.73 | -58.03 | -79.16 | -84.15 |
| | | Medium | 21 | 8.33 | 17.73 | 104.72 | 10.23 | -8.16 | 16.00 | 18.41 |
| | | Good | 19 | 11.16 | 17.43 | 217.49 | 14.75 | -0.09 | -28.61 | -24.68 |
| | | $\Delta$ (max−min) | 50 | 3.46 | 16.06 | 112.77 | 4.52 | 57.94 | 95.16 | 102.56 |
| SigLIP2 | combined | Poor | 10 | 7.70 | 33.48 | 115.11 | 10.73 | -58.02 | -81.27 | -79.27 |
| | | Medium | 21 | 8.33 | 17.73 | 104.72 | 10.23 | -8.16 | 11.13 | 13.62 |
| | | Good | 19 | 11.16 | 17.43 | 217.49 | 14.75 | -0.09 | -29.85 | -29.60 |
| | | $\Delta$ (max−min) | 50 | 3.46 | 16.05 | 112.77 | 4.52 | 57.93 | 92.40 | 92.89 |

Table 17: Pediatric subgroup analysis by sex with per-group raw metrics and a $\Delta$ (max−min) on EchoNet-Pediatric

| Model | View | Group | $n$ | MAE | NMAE (%) | MSE | RMSE | $R^2$ (%) | Pearson $r$ (%) | Spearman $\rho$ (%) |
|---|---|---|---|---|---|---|---|---|---|---|
| EchoCLIP | combined | F | 301 | 14.39 | 27.33 | 316.63 | 17.79 | -462.00 | 18.86 | 19.04 |
| | | M | 355 | 13.32 | 21.99 | 276.60 | 16.63 | -241.52 | 32.48 | 13.21 |
| | | $\Delta$ (max−min) | 656 | 1.07 | 5.34 | 40.03 | 1.16 | 220.48 | 13.62 | 5.83 |
| EchoPrime | combined | F | 301 | 4.86 | 9.22 | 45.92 | 6.78 | 18.49 | 45.49 | 32.88 |
| | | M | 355 | 5.95 | 9.83 | 64.43 | 8.03 | 20.45 | 45.76 | 23.71 |
| | | $\Delta$ (max−min) | 656 | 1.09 | 0.61 | 18.51 | 1.25 | 1.96 | 0.27 | 9.17 |
| PanEcho | combined | F | 301 | 8.68 | 16.49 | 139.82 | 11.82 | -148.18 | 44.58 | 39.47 |
| | | M | 355 | 9.50 | 15.69 | 158.11 | 12.57 | -95.22 | 49.02 | 30.58 |
| | | $\Delta$ (max−min) | 656 | 0.82 | 0.80 | 18.29 | 0.75 | 52.96 | 4.44 | 8.89 |
| BioMedCLIP | combined | F | 301 | 18.34 | 34.83 | 373.58 | 19.33 | -563.08 | 11.84 | 7.70 |
| | | M | 355 | 18.32 | 30.26 | 385.46 | 19.63 | -375.93 | 9.41 | 6.90 |
| | | $\Delta$ (max−min) | 656 | 0.02 | 4.57 | 11.88 | 0.30 | 187.15 | 2.43 | 0.80 |
| DINOv3 | combined | F | 301 | 18.48 | 35.11 | 367.06 | 19.16 | -551.53 | -1.86 | -2.74 |
| | | M | 355 | 18.07 | 29.86 | 361.42 | 19.01 | -346.25 | -0.23 | -7.48 |
| | | $\Delta$ (max−min) | 656 | 0.41 | 5.25 | 5.64 | 0.15 | 205.28 | 1.63 | 4.74 |
| SigLIP2 | combined | F | 301 | 18.46 | 35.07 | 366.38 | 19.14 | -550.31 | -3.66 | 1.93 |
| | | M | 355 | 18.06 | 29.83 | 360.76 | 18.99 | -345.44 | -12.81 | -3.91 |
| | | $\Delta$ (max−min) | 656 | 0.40 | 5.24 | 5.62 | 0.15 | 204.87 | 9.15 | 5.84 |

Table 18: Pediatric subgroup analysis by age bin with per-group raw metrics and a $\Delta$ (max−min) on EchoNet-Pediatric.

| Model | View | Group | $n$ | MAE | NMAE (%) | MSE | RMSE | $R^2$ (%) | Pearson $r$ (%) | Spearman $\rho$ (%) |
|---|---|---|---|---|---|---|---|---|---|---|
| EchoCLIP | combined | 0–1 | 55 | 11.99 | 22.87 | 216.65 | 14.72 | -73.40 | 12.47 | 10.48 |
| | | 1–5 | 121 | 15.23 | 65.52 | 324.21 | 18.01 | -1352.36 | 22.38 | 15.09 |
| | | 6–12 | 242 | 14.56 | 23.97 | 324.38 | 18.01 | -257.10 | 33.82 | 17.91 |
| | | 13–18 | 240 | 12.73 | 32.10 | 267.30 | 16.35 | -388.52 | 28.88 | 18.26 |
| | | $\Delta$ (max−min) | 658 | 3.24 | 42.65 | 107.73 | 3.29 | 1278.96 | 21.35 | 7.78 |
| EchoPrime | combined | 0–1 | 55 | 7.23 | 13.79 | 93.83 | 9.69 | 24.90 | 55.14 | 32.88 |
| | | 1–5 | 121 | 5.02 | 21.62 | 38.78 | 6.23 | -73.72 | 21.16 | 11.28 |
| | | 6–12 | 242 | 5.52 | 9.09 | 66.42 | 8.15 | 26.89 | 52.03 | 30.16 |
| | | 13–18 | 240 | 5.17 | 13.03 | 44.97 | 6.71 | 17.81 | 44.07 | 31.70 |
| | | $\Delta$ (max−min) | 658 | 2.21 | 12.53 | 55.05 | 3.46 | 100.61 | 33.98 | 21.60 |
| PanEcho | combined | 0–1 | 55 | 9.82 | 18.73 | 174.99 | 13.23 | -40.06 | 50.22 | 48.54 |
| | | 1–5 | 121 | 9.81 | 42.23 | 167.68 | 12.95 | -651.14 | 39.05 | 34.40 |
| | | 6–12 | 242 | 9.84 | 16.20 | 174.39 | 13.21 | -91.98 | 50.41 | 28.56 |
| | | 13–18 | 240 | 7.84 | 19.77 | 108.83 | 10.43 | -98.89 | 46.62 | 35.39 |
| | | $\Delta$ (max−min) | 658 | 2.00 | 26.03 | 66.16 | 2.80 | 611.08 | 11.36 | 19.98 |
| BioMedCLIP | combined | 0–1 | 55 | 17.21 | 32.82 | 355.68 | 18.86 | -184.68 | 20.90 | 16.92 |
| | | 1–5 | 121 | 19.80 | 85.18 | 428.21 | 20.69 | -1818.25 | -6.12 | -7.14 |
| | | 6–12 | 242 | 18.49 | 30.45 | 385.74 | 19.64 | -324.65 | 9.02 | 0.68 |
| | | 13–18 | 240 | 17.60 | 44.39 | 353.11 | 18.79 | -545.34 | 12.23 | 14.48 |
| | | $\Delta$ (max−min) | 658 | 2.59 | 54.73 | 75.10 | 1.90 | 1633.57 | 27.02 | 24.06 |
| DINOv3 | combined | 0–1 | 55 | 17.88 | 34.08 | 356.39 | 18.88 | -185.24 | -3.36 | -13.69 |
| | | 1–5 | 121 | 19.87 | 85.51 | 417.26 | 20.43 | -1769.21 | -1.10 | -4.61 |
| | | 6–12 | 242 | 18.46 | 30.40 | 369.31 | 19.22 | -306.56 | -0.02 | -6.33 |
| | | 13–18 | 240 | 17.27 | 43.57 | 331.58 | 18.21 | -506.00 | -0.96 | -3.08 |
| | | $\Delta$ (max−min) | 658 | 2.60 | 55.11 | 85.68 | 2.22 | 1583.97 | 3.34 | 10.61 |
| SigLIP2 | combined | 0–1 | 55 | 17.86 | 34.05 | 355.77 | 18.86 | -184.75 | 13.13 | 2.56 |
| | | 1–5 | 121 | 19.85 | 85.43 | 416.49 | 20.41 | -1765.74 | -11.94 | -10.97 |
| | | 6–12 | 242 | 18.45 | 30.37 | 368.66 | 19.20 | -305.84 | -16.46 | 1.84 |
| | | 13–18 | 240 | 17.26 | 43.53 | 330.95 | 18.19 | -504.84 | -7.65 | -2.36 |
| | | $\Delta$ (max−min) | 658 | 2.59 | 55.06 | 85.54 | 2.22 | 1580.99 | 29.59 | 13.53 |

Table 19: Pediatric subgroup analysis by BMI bin with per-group raw metrics and a $\Delta$ (max$-$min) on EchoNet-Pediatric.

| Model | View | Group | $n$ | MAE | NMAE (%) | MSE | RMSE | $R^2$ (%) | Pearson $r$ (%) | Spearman $\rho$ (%) |
|---|---|---|---|---|---|---|---|---|---|---|
| EchoCLIP | combined | Low | 345 | 14.20 | 23.39 | 310.80 | 17.63 | -209.39 | 30.32 | 21.03 |
| | | Healthy | 205 | 13.77 | 39.68 | 287.43 | 16.95 | -786.11 | 16.62 | 2.60 |
| | | High | 107 | 12.57 | 41.78 | 257.20 | 16.04 | -624.20 | 22.98 | 16.04 |
| | | $\Delta$ (max$-$min) | 657 | 1.63 | 18.39 | 53.60 | 1.59 | 576.72 | 13.70 | 18.43 |
| EchoPrime | combined | Low | 345 | 5.96 | 9.82 | 70.96 | 8.42 | 29.37 | 54.50 | 35.39 |
| | | Healthy | 205 | 4.97 | 14.31 | 41.53 | 6.44 | -28.05 | 9.67 | 5.84 |
| | | High | 107 | 4.64 | 15.40 | 34.00 | 5.83 | 4.27 | 42.18 | 37.61 |
| | | $\Delta$ (max$-$min) | 657 | 1.32 | 5.58 | 36.96 | 2.59 | 57.42 | 44.83 | 31.77 |
| PanEcho | combined | Low | 345 | 10.41 | 17.14 | 190.37 | 13.80 | -89.50 | 51.27 | 39.42 |
| | | Healthy | 205 | 7.40 | 21.31 | 100.68 | 10.03 | -210.37 | 29.31 | 23.83 |
| | | High | 107 | 8.17 | 27.13 | 110.51 | 10.51 | -211.16 | 35.44 | 34.95 |
| | | $\Delta$ (max$-$min) | 657 | 3.01 | 9.99 | 89.69 | 3.77 | 121.66 | 21.96 | 15.59 |
| BioMedCLIP | combined | Low | 345 | 18.40 | 30.29 | 385.16 | 19.63 | -283.40 | 9.24 | 5.04 |
| | | Healthy | 205 | 18.38 | 52.96 | 378.86 | 19.46 | -1067.97 | 9.05 | 2.05 |
| | | High | 107 | 17.93 | 59.57 | 363.25 | 19.06 | -922.79 | 16.51 | 17.64 |
| | | $\Delta$ (max$-$min) | 657 | 0.47 | 29.28 | 21.91 | 0.57 | 784.57 | 7.46 | 15.59 |
| DINOv3 | combined | Low | 345 | 18.18 | 29.94 | 362.89 | 19.05 | -261.23 | -2.27 | -10.90 |
| | | Healthy | 205 | 18.36 | 52.90 | 364.33 | 19.09 | -1023.18 | -2.51 | -1.71 |
| | | High | 107 | 18.32 | 60.88 | 365.72 | 19.12 | -929.76 | 6.96 | 7.37 |
| | | $\Delta$ (max$-$min) | 657 | 0.18 | 30.94 | 2.83 | 0.07 | 761.95 | 9.47 | 18.27 |
| SigLIP2 | combined | Low | 345 | 18.16 | 29.91 | 362.25 | 19.03 | -260.60 | -15.33 | -7.31 |
| | | Healthy | 205 | 18.34 | 52.84 | 363.60 | 19.07 | -1020.93 | 2.17 | 4.39 |
| | | High | 107 | 18.31 | 60.82 | 365.07 | 19.11 | -927.92 | 6.62 | 10.71 |
| | | $\Delta$ (max$-$min) | 657 | 0.18 | 30.91 | 2.82 | 0.08 | 760.33 | 21.95 | 18.02 |