# OpenReview forum: "CardioBench: Do Echocardiography Foundation Models Generalize Beyond the Lab?"
_ICLR.cc/2026/Conference — ICLR 2026 Conference Withdrawn Submission_

### Official Review · Reviewer_pWHE · 2025-10-25

**Soundness:** 3
**Presentation:** 2
**Contribution:** 2
**Rating:** 2
**Confidence:** 4

**Summary:**

CardioBench addresses the pain point of "lack of cross-task and cross-dataset comparability in echocardiography FM" by proposing a unified evaluation and several insightful analyses (time series, search, text axis, and physiological significance)—a direction worthy of encouragement. However, if it claims to be a "standardized benchmark," it still needs to strengthen protocol transparency, statistical rigor, track fairness (external search), and engineering integrity (prompts/thresholds/units). We recommend focusing on experiments and documentation in small sample/PEFT tracks, public search libraries, units and processing rules, and statistical robustness. This will significantly enhance the persuasiveness of the paper.

**Strengths:**

1. Broad Coverage, Unified Protocols for Public Benchmarks

This standardizes eight public datasets, covering four regression and five classification tasks. It provides standardized preprocessing and data partitioning, and evaluates under consistent zero-shot, probing, and alignment protocols, facilitating fair reproduction and horizontal comparison. This represents a substantial improvement over the current fragmented evaluation landscape.

2. From Results to Actionable Insights

In addition to reporting scores, this paper systematically analyzes the complementary strengths and limitations of temporal modeling, text encoders, retrieval-based methods, and general-purpose large models in echocardiography tasks. It also characterizes representation quality using four metrics: task performance, cluster consistency, cross-modal alignment, and demographic robustness, providing clear guidance for subsequent model design.

**Weaknesses:**

1. The benchmark's "added value" is insufficiently distinguished from prior work such as ETAB.
The paper argues that CardioBench unifies eight public datasets, covering four regression and five classification tasks, and provides standardized preprocessing/partitioning and unified evaluation scripts. A clearer explanation is needed regarding what CardioBench adds, what changes, and what biases it avoids compared to existing hyper-benchmarks such as ETAB (e.g., task overlap, overfitting to a single perspective, or greater friendliness to time series/retrieval models). The current comparison and motivational explanations are somewhat conceptual, lacking a one-to-one mapping/difference table at the task level and empirical discussion of why this design is more clinically relevant.

2. Biased selection of evaluation protocols may affect the generalizability of conclusions.
The authors deliberately excluded fine-tuning and small-shot training, focusing solely on zero-shot and linear probing, reasoning that "few samples can easily lead to overfitting and instability." However, real-world implementations often require adaptation with minimal annotations. Comparing only under zero-shot and linear probing yields uneven benefits across architectures (for example, models with explicit retrieval or explicit view heads perform better under zero-shot conditions), thus failing to represent actual deployable capabilities. We recommend supplementing with baselines and sensitivity analyses using a unified budget for small-shot, distillation, and parameter-efficient fine-tuning (PEFT/LoRA).

3. Insufficient details regarding metrics, units, and protocols hinder comparability. While the regression uses MAE, the units and dimensions of EF, LVIDd, IVSd, and LVPWd are not clearly defined in the main text (e.g., mm/cm? Are they voxel-normalized or measured using a unified caliber?). Has dimensional alignment and dimensional uncertainty propagation been performed across datasets? While macro-F1 is a reasonable categorization, the numerous missing values ​​("-") and the different task combinations in the tables make horizontal comparison difficult. It is recommended that unified protocols for units, measurement rules, interpolation/resampling, frame length cropping, and cardiac cycle alignment be provided in the main text or an appendix.

4. The "missing items and outliers" in Tables 1 and 2 are questionable and require more rigorous reporting of completeness and robustness. For example, some models in Table 3 have F1 values ​​≈ 0 on several datasets (e.g., DINOv3 has an F1 value of 0 on multiple datasets). This suggests engineering issues related to label mapping, class coverage, thresholds, or cue terms, rather than inherent model capabilities. Explanations are needed for view name mapping, class imbalance handling, decision thresholds, and failure examples. The mean ± variance of multiple random seeds and statistical significance should also be provided.

5. Fairness and Reproducibility of "Search-Based Models + Private Repositories"
The paper acknowledges EchoPrime's reliance on private repositories for retrieval and suggests that similarity may drive performance differences on certain datasets, potentially leading to stronger or weaker performance. This constitutes external resource differences within a unified benchmark, undermining the claim of "reproducibility and fair comparison." Recommendations: Provide alternative public retrieval repositories or retest under a "ban external private repositories" setting; and clearly indicate tracks that allow or disallow external knowledge.

6. Consistency in naming and formatting: The capitalization of "SigLIP2/SigLip2" is inconsistent in the text; the column annotations (blue/green) in Tables 1/2 are not user-friendly when printed in black and white; it is recommended to add a table of task abbreviations and clearer descriptions of vacancy marks.

7. Figure 2: Similarity matrix diagram: It is recommended to provide frame sampling rate, sequence length, and normalization details to avoid "similarity bias caused by sampling method".

**Questions:**

1. How are the units/dimensions of each regression metric aligned with those across datasets? Are they standardized to mm or cm? Is the EF standardized as a percentage with consistent rounding/decimal placement?

2. Are the class sets and mapping vocabularies for view classification publicly available? Are the several 0.00 scores in Table 3 due to uncovered classes/threshold setting issues?

3. Can EchoPrime's search evaluation reproduce Tables 1/2 without accessing private repositories? If so, are the index size/composition and similarity metrics publicly available?

4. Is the full set of prompts, thresholds, and post-processing for Zero-shot/Probe fully documented in Appendix D/E and released with the code?

5. Have the demographic stratifications been tested for significance and effect size? Have other tasks (e.g., AS/PAH/STEMI, views) also been audited for bias?

---

### Official Review · Reviewer_915v · 2025-10-31

**Soundness:** 3
**Presentation:** 3
**Contribution:** 2
**Rating:** 4
**Confidence:** 4

**Summary:**

This paper introduces CardioBench, a comprehensive, publicly available benchmark designed to standardize the evaluation of foundation models for echocardiography. The authors address a gap in the field, where existing models are often evaluated on private datasets, hindering fair comparison and reproducibility. CardioBench unifies eight distinct, publicly available datasets, covering a clinically relevant set of nine tasks. The authors evaluate a range of foundation models. Using standardized zero-shot and linear probing protocols, the study reveals that no single model dominates. The paper provides insights into model biases and the limitations of current architectures, offering a reproducible platform to guide future research.

**Strengths:**

- The paper addresses a clear need within the medical AI community. CardioBench provides a well-structured and comprehensive solution to the lack of a standardized and public benchmark for fair comparison.

- The benchmark is thorough. It unifies eight public datasets to cover a wide spectrum of clinically relevant tasks.

- The authors commit to releasing all preprocessing code, standardized data splits, and evaluation pipelines. This makes CardioBench useful and have high reproducibility.

**Weaknesses:**

- It would be better if the author could provide more practical insights into how the foundation models address the mentioned challenges, e.g., by trying some methods and providing result comparisons of fully fine-tuning (or parameter-efficient fine-tuning) existing foundation models, or by proposing their own architecture to alleviate current problems.

- The paper excels at identifying which models perform well but is weaker on explaining why they do so. A deeper dive into which architectural components or pretraining objectives specifically enable this superior performance would have strengthened the paper’s conclusions.

- Why is EchoFM included in the linear probing results (Table 2) but absent from the zero-shot results (Table 1)?

**Questions:**

Please see Weakness part.

---

### Official Review · Reviewer_n1i8 · 2025-11-01

**Soundness:** 3
**Presentation:** 3
**Contribution:** 1
**Rating:** 2
**Confidence:** 3

**Summary:**

In order to evaluate the newly introduced models' echocardiography-specific FMs, the author introduces a standard benchmark for echocardiography foundation models that combine eight public datasets across four regression and five classification tasks with consistent zero-shot, probing, and alignment protocols. Results show that no single model is consistently dominant across all tasks, all datasets, and all evaluation settings. Temporal modeling is critical for functional tasks, retrieval augmentation enhances robustness when there is a distribution shift, and domain cardiac text encoders can encode physiologic structure but are unstable across datasets. General-purpose encoders such as SigLIP2 transfer well on coarse classification and after light probing, but lag on fine-grained measurements. View recognition favors supervised heads, and embeddings rarely form clean global clusters. In addition, the subgroup analysis reveals datasets’ biases, inspires building hybrid systems that combine temporal modeling, retrieval, and light, structured supervision, with standard reporting of each subgroup’s performance.

**Strengths:**

1. Colorful and clear diagrams that are helpful for comprehension.
2. This paper tests all approaches on the same datasets with the same evaluation protocol; the study prevents the incomparability caused by prior work running on separate datasets and metrics.
3. This paper is well organized and reads smoothly.
4. This paper introduces CardioBench, which unifies eight public datasets into a single standardized evaluation.

**Weaknesses:**

1. The paper has limited novelty, the core contribution is a standardized benchmark unifying public echo datasets. However, ETAB (https://proceedings.neurips.cc/paper_files/paper/2022/hash/796501434d0dc3a039d5b91261f7f889-Abstract-Datasets_and_Benchmarks.html) already introduced a public dataset, a unified evaluation suite for echocardiography representation learning with code and protocol. CardioBench mainly updates coverage to recent FMs rather than creating a fundamentally new evaluation paradigm.

2. CardioBench corroborates but does not materially move these insights forward. The claims about temporal modeling are critical, retrieval aids OOD robustness, and view recognition benefits from supervised heads largely echo prior FM work such as EchoCLIP, EchoPrime, and EchoFM. While these sources are cited in the paper, the work largely reaffirms their conclusions without introducing new methodology or previously unknown insights.

3. The paper’s bias analysis is limited to descriptive subgroup comparisons and boxplots, primarily for EF regression, which constrains what we can conclude about fairness. It does not provide threshold-independent metrics for classification, such as AUROC or AUPRC. Moreover, fairness gaps at clinically chosen operating points. For instance, TPR/FPR gaps are omitted, and calibration checks like ECE and Brier are crucial when applying a single threshold across groups. Several subgroups appear small, so variance and CIs are not quantified, raising the risk of noisy or non-reproducible differences; multiple subgroup slices are tested without clear correction for multiple comparisons, inviting false positives. The observed gaps may reflect acquisition bias rather than model bias. Finally, the analysis focuses on averages and medians, underreporting tail risk that matters most for safety-critical deployment.

4. The study stops short of explaining why the observed patterns arise. There is no controlled, domain aware audit of the retrieval gallery, such as leakage checks or index selection across sites and devices. It offers no mechanism level analysis of thresholding and calibration for fairness, such as subgroup ECE, Brier score, and operating point sensitivity. It also provides no principled link between embedding geometry and downstream error, beyond UMAP and ARI. Finally, the paper does not explore scaling behavior across data size, model size, and temporal resolution, which would clarify whether the reported effects are capacity limited or data limited.

**Questions:**

See weaknesses.

---

### Note · Authors · 2025-11-19

I have read and agree with the venue's withdrawal policy on behalf of myself and my co-authors.